# Florida Keys *Cassiopea* host benthos-like external microbiomes and a gut dominated by *Vibrio*, *Endozoicomonas* and *Mycoplasma*

Kaden M. Muffett[1]*, Jessica M. Labonté[2], Maria Pia Miglietta[2]

**1** University of California Merced, Merced, California, United States of America, **2** Texas A&M University at Galveston, Galveston, Texas, United States of America

* kmmuffett@gmail.com

## Abstract

Interactions with microbial communities fundamentally shape metazoans' physiology, development, and health across marine ecosystems. This is especially true in zooxanthellate (symbiotic algae-containing) cnidarians. In photosymbiotic anthozoans (e.g., shallow water anemones and corals), the key members of the associated microbiota are increasingly well studied, however there is limited data on photosymbiotic scyphozoans (true jellyfish). Using 16S rRNA barcoding, we sampled the internal and external mucus of the zooxanthellate Upside-Down Jellyfish, *Cassiopea xamachana* during August throughout eight sites covering the full length of the Florida Keys. We find that across sites, these medusae have low-diversity internal microbiomes distinct from the communities of their external surfaces and their environment. These internal communities are dominated by only three taxa: *Endozoicomonas* cf. *atrinae*, an uncultured novel *Mycoplasma*, and *Vibrio* cf. *coralliilyticus*. In addition, we find that *Cassiopea* bell mucosal samples were high diversity and conform largely to the communities of surrounding sediment with the addition of *Endozoicomonas* cf. *atrinae*. The microbial taxa we identify associated with wild Florida Keys *Cassiopea* bear a strong resemblance to those found within photosymbiotic anthozoans, increasing the known links in ecological position between these groups.

## Introduction

Cnidarian-photosymbiont interactions are central to ecosystem function in coral reefs and beyond. Algal photosymbionts, or zooxanthellae, are present in some well-known stony corals, octocorals, and anemones, but also play important roles in a variety of lesser-known jellyfishes, chief among them, jellyfish of the genus *Cassiopea* [1–4]. For symbiotic anthozoans, microbial communities play a key role in maintaining health of this photosymbiont partnership, conferring disease resistance and heat tolerance on hosts [5–7]. These microbial communities are often specialized, with

**Data availability statement:** All raw sequencing files are available from the SRA database (BioProject accession numbers PRJNA1020446, PRJNA1020388.)

**Funding:** This work was funded by Texas Sea Grant # NA18OAR4170088, NSF EAGER award # 1936565, and Texas A&M University Galveston. The funders had no role in study design, data collection and analysis, decision to publish, or preparation of the manuscript.

**Competing interests:** The authors have declared that no competing interests exist.

distinct corals demonstrating co-evolution with members of their bacterial communities [8]. Several prevalent genera, such as *Endozoicomonas*, are found across the microbiomes of many hosts [9]. While anthozoan communities are under active investigation, few analyses have documented the stability and nature of microbiome communities associated with zooxanthellate jellyfishes, a group with divergent life history traits from Anthozoa but with a comparable obligate symbiosis with dinoflagellates (family Symbiodiniaceae).

The Upside-Down Jellyfish, *Cassiopea* spp., is a model organism for cnidarian-photosymbiont interactions [10,11]. These medusae are residents of tropical and subtropical waters globally and can survive in water that is frequently shallow (as little as 8 cm depth) and warm (up to 33˚C for *Cassiopea andromeda* (Forskal, 1775)) [12,13]. *Cassiopea* obligatorily hosts algal symbionts, losing mass even when well fed while in aposymbiotic conditions [14]. As in corals and other scyphozoans, mucus plays a central role for *Cassiopea*, presenting an interface and barrier between the host tissue and the marine environment [15,16]. In contrast to other symbiotic cnidarians, such as photosymbiotic stony corals, *Cassiopea* can tolerate large temperature ranges (18C to 33C) and intense temperature fluctuations (+/- 6C) [17–19]. Understanding the *Cassiopea* natural microbial community could provide new potential insights into the function and diversity of cnidarian holobionts.

Past jellyfish (class Scyphozoa) microbial studies have focused on the microbiome of lab-reared specimens [20,21] or limited sample sizes [22–24]. While these types of studies may be able to track state changes across life stages [20] and between condition types (symbiotic/aposymbiotic) [21], there is little guarantee that the microbial communities of jellyfish kept under prolonged laboratory conditions translate bear resemblance to *in situ* microbiomes. Recent *in situ* studies have provided insights into microbial taxa associated with some scyphozoans ("true jellyfish") but have not included species of the genus *Cassiopea*. In the moon jellyfish, *Aurelia* spp*.,* individuals from the Chinese coast were *Vibrionaceae*-dominated, while samples from the northwestern Atlantic were *Mycoplasma*-dominated [24,25]. Between the Indonesian marine lakes Kakaban, Haji Buang, and Tanah Bamban, *Mastigias* sp. microbial community composition varied significantly by lake, but consistently included *Oceanospirillales* (primarily *Endozoicomonas*-like) [22]. Across hosts, *Mycoplasma*-like bacteria are common components of the identified gut and tissue microbiome of jellyfishes, though their functions remains unknown [24–29]. Vertebrate pathogen research is often a focus of scyphozoan microbiome discovery—*Tenacibaculum* has been identified in several scyphozoan species, e.g., *Cotyllorhiza tuberculata, Aurelia aurita,* and *Pelagia noctiluca*, and, as a fish pathogen, which can have important fisheries ramifications [24,25,27,30].

As most studied jellyfish populations occur in coastal waters only sporadically, the relative consistency of *Cassiopea* assemblages provides a system in which body compartment microbiomes can be studied without the confounding impacts of single source sites. *Cassiopea* spp. presents an opportunity to act as a comparison group for the microbial communities associated with both other scyphozoans and symbiotic anthozoans. The ubiquity of *Cassiopea* assemblages within the Florida Keys allows

for the examination of differences in the microbial community of one population (Florida Keys *Cassiopea*) at one time point across a broad geographic area.

The aim of this work is to undertake a first exploratory study of the wild microbiome of *Cassiopea*, and to identify the core components of the internal and external mucosal microbiomes of *Cassiopea* within the Florida Keys.

## Materials and methods

### Sampling

A total of 55 *Cassiopea* medusae were collected by wading in eight sites along the length of the Florida Keys in late August 2021 over eight days (see Table 1 for a list of sites). Species identification for each individual is reported in Muffett & Miglietta 2023 and summarized in Supplementary Table S1 in S1 File [31]. At each site, coordinates, temperature, time of day, depth, salinity, and pH were recorded (Table 1). Salinity was measured with a refractometer, and temperature and pH were measured with an Apera A1209. A minimum of two and a maximum of 10 medusae were sampled per site. At each site and in between each medusa, all equipment was cleaned with absolute ethanol and 10% bleach solution. A 1 L water sample was collected from above (>10 cm above the benthos) each medusa aggregation and filtered (0.2 um pore size, Thermo Scientific Cat No.09-740-30G) with a manual sampler as described in the USGS manual water sampling protocol [32], replicate filters were placed in 15 mL of dimethyl sulfoxide ethylenediamine tetraacetic acid saturated salt storage solution (1 L pH 7.5: 93.06 g EDTA, 60 mL 20% NaOH solution, 20 mL 25% HCl, 40 mL DMSO, 800 mL water, NaCl to oversaturation; see ref. 30) (commonly referred to as DESS), one filter in each buffer. At each *Cassiopea* medusa

**Table 1. List of collection sites.** Table with Site Name (the collection location), site code (the shorthand used for these locations named for local landmarks, see supplementary table S1 in S1 File), latitude, longitude, salinity (parts per thousand), pH, site depth in meters, surface temperature (in Celsius), average diameter of collected medusae from the site (centimeters), and date of collection with time of day (AM/PM). Number of paired gastrovascular cavity (GVC) and bell swab samples per collection are for *Cassiopea xamachana* (CX). For collections that included *C. andromeda* mitotyped medusae, the number of *C. andromeda* samples is presented after the dash (CA). These values only include samples collected in DESS. Due to a storm warning, collections from Bahia Honda occurred across two days (TP1, TP2).

| Site Name | Site code | Latitude | Longitude | Salinity (ppt) | pH | Temp (ºC) | Average depth (m) | Average col. medusa diam (cm) | # of GVC Samps | # of Bell Samps | Date (AM/PM) |
|---|---|---|---|---|---|---|---|---|---|---|---|
| Key West | GB | 24.561526 | −81.7881727 | 36 | 8.2 | 30.4 | 0.56 | 6.62 | CX 3/ CA 1* | CX 3/ CA 1* | 15.8.2021 PM |
| Cudjoe Key | CK | 24.6775378 | −81.4991819 | 36 | 8.4 | 32.6 | 0.36 | 5.91 | CX 4/ CA 2* | CX 4/ CA 2* | 17.8.2021 AM |
| Big Pine Key | BPK | 24.6978731 | −81.3572574 | 35 | 7.7 | 29.1 | 0.15 | 4.43 | CX 4 | CX 4 | 15.8.2021 PM |
| Bahia Honda Key TP1 | VP | 24.6825793 | −81.2293673 | 37 | 8.3 | 31.8 | 0.86 | 12.41 | CX 2 | CX 2 | 13.8.2021 PM |
| Bahia Honda Key TP2 | VP | 24.6825793 | −81.2293673 | 36 | 8 | 29.4 | 0.51 | 10.28 | CX 2 | CX 2 | 15.8.2021 AM |
| Marathon Key | MK | 24.69396 | −81.0980507 | 31 | 8.1 | 35.2 | 0.08 | 6.53 | CX 7 | CX 7 | 11.8.2021 PM |
| Lower Matacumbe Key | LW | 24.8582157 | −80.7267927 | 35 | 8.1 | 32.9 | 0.89 | 11.40 | CX 2 | CX 2 | 16.8.2021 PM |
| Tavernier | HHP | 25.0233783 | −80.4940316 | 31 | 8.1 | 33.3 | 0.35 | 12.26 | CX 3 | CX 3 | 16.8.2021 AM |
| Key Largo | KL | 25.0872104 | −80.4415797 | 35 | 7.8 | 32.0 | 0.56 | 4.30 | CX 3/ CA 1* | CX 3/ CA 1* | 18.8.2021 AM |

aggregation, duplicate scrapings of ~10 g of sediment collections (substrate sample) were placed in 15 mL of DESS storage solution.

Each *Cassiopea* medusa was collected from shallow waters <1.5 m within a single assemblage at each site. Medusae were gently grasped by the oral arms, lifted from the water, swabbed along the apex of the bell, then placed bell-down onto a pre-prepared dissection plate (Fig 1). External bell swabs were done in replicate and placed in 3 mL DESS. Medusa oral arm base was bisected between oral arm groups, and replicate swabs were inserted and circled the digestive cavity (hereafter referred to as gastrovascular cavity or GVC). Medusae<3 cm in diameter were not sampled for gastrovascular cavity and bell microbiome. A total of 46 medusae were sampled. One internal and one external swab from each medusa was placed in DESS. A section of bell tissue was then collected and placed in ethanol for mitochondrial species confirmation (see 28). The size of each medusa was measured, and individual medusae were photographed. As sampling was done in a nonsterile environment, blanks for buffer with swab and buffer with filter were also taken.

## DNA extraction and sequencing

Samples in DESS were maintained at room temperature for three to ten days and then frozen. DESS can be stored at room temperature for at least thirty days without significantly compromising community integrity [33]. DNA from all samples (see Supplementary Table S1 in S1 File) was then extracted using Zymobiomics DNA Kits (PN# D4300T). All samples were pre-tested through the amplification of the 16S rRNA gene (variable region V4). Only samples that produced visible bands for the V4 region were sent for 16S rRNA V3-V4 preparation and sequencing, with a maximum of 7 specimens per site. In total, the DNA from 34 paired medusa internal and external microbiomes were sent for sequencing.

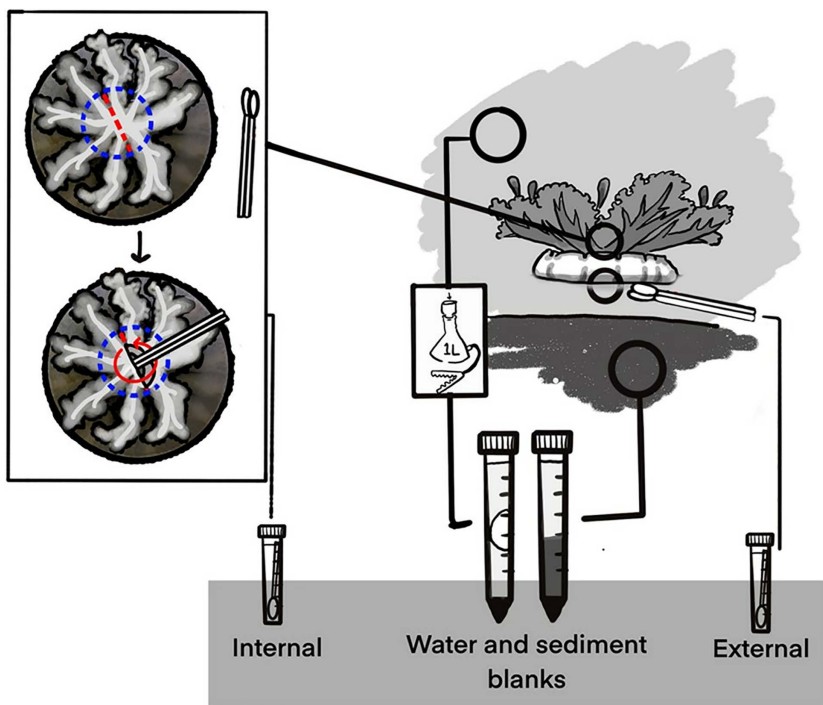

**Fig 1. Sampling strategy for medusae collection onsite.** Demonstration of the typical orientation of medusa (right upper) and swab locations (apical bell surface and internal to medusa). Gastrovascular cavity (GVC) access strategy is displayed on the left and relevant environmental samples (water filter, substrate sample) are displayed on the right.

Sequencing was performed by the University of Texas Medical Branch Sequencing Lab. Library was prepared with Zymo Quick 16S V3-V4 kit (PN #D6400) in 20 uL reactions (10 uL Quick-16S qPCR premix, 4 uL Quick-16S V3-V4 primers (341f: (CCTACGGGDGGCWGCAG, CCTAYGGGGYGCWGCAG) and 806r: (GACTACNVGGGTMTCTAATCC)), 4 uL water and 2 uL sample). qPCR was run using Roche LightCycler 480 Instrument II (10 min 95C, [30sec 95C, 30sec 55C, 3 min 72C]x20). PCRs were enzymatically cleaned according to Zymo Quick-16S kit protocols, and barcodes were added in a secondary reaction at 20 uL (10 min 95˚C, [30 sec 95˚C, 30 sec 55˚C, 3 min 72˚C]x5). Plates included two replicate microbial community standards (Zymobiomics Cat. No. D6300) and two PCR negative controls. Samples were loaded at 10 pM, with 15% 10 pM PhiX sequencing control, and run on Illumina MiSeq with a 600 cycle v3 kit.

## Data preprocessing

Data generated was run through the DADA2 v 1.26 pipeline on R v 4.4 with only limited deviations from the standard protocols for amplicon sequence variant (ASV) generation [34,35]. Briefly, sequences were filtered and trimmed ("filterAndTrim(truncLen=c(285,230), maxN=0, maxEE=c(4,4), truncQ=2,trimLeft = c(17,21)"), errors were error corrected ("learnErrors", "dada"), and sequences were merged ("mergePairs"). Reads between 380 and 458 bp were retained. Chimeric reads were removed ("removeBimeraDenovo"), then taxonomy was assigned using the Silva v 138.1 database ("assignTaxonomy") [36]. Metrics on number of sequences passing quality control along with the full code can be found on the github (https://github.com/kadenmuffett/Cassiopea-Code-Microbiome-Repository). Sequence and metadata tables were imported into phyloseq v 1.42, where reads assigned to mitochondria and chloroplast were removed. Contaminants were removed using the decontam package v. 1.18, identifying and removing 47 amplicon sequence variants based on differential abundance in swab and buffer controls [37,38].

Based on host genotyping, samples not belonging to *Cassiopea xamachana* Bigelow, 1892 were removed. The dataset included 60 paired bell and GVC samples (30 of each), 8 water samples and 8 substrate samples with 6.47 mil sequences across 34,808 taxa. The complete dataset (NCBI Bioproject Accession number PRJNA1020388) contains 12.78 million reads.

## Statistical analyses

Rarefaction curves were generated in vegan 2.6.6.1 ("rarecurve") (Sfig. 1) [39]. Shannon diversity was computed with vegan and observed diversity was ploted with phyloseq ("estimate_richness", "plot_richness") [37]. In order to identify the degree to which diversity spanned primarily shallow or deep phylogenetic distances, Faith's phylogenetic diversity was calculated for all samples and sample types. Faith's phylogenetic diversity was computed using the MiscMetabar v 0.10.1, phanghorn v 2.12.1, metagMisc v 0.5.0 and PhyloseqMeasures v 2.1 packages by first generating an ML tree from all sequences over 300 reads across the dataset, then computing per sample Faith's diversity [40–45]. Faith's diversity was displayed with ggstatsplot v 0.13.0 [46]. Bray-Curtis distance was computed in phyloseq with "distance" to "ordinate" to "plot_ordination". Between group comparisons were completed with the pairwiseAdonis package v 0.4.1 (" pairwise. adonis2") on Bray-Curtis distance measurements [47]. Statistical comparisons were computed with Kruskal-Wallis using the Kruskal.test and pairwise comparisons using the "kruskal.test" and "dunn.test" commands with holm p adjustment [48].

Data transmutation for starplots and display was assisted by the speedyseq package v 0.5.3.9021 [49]. Star plots were made using the stars package v 0.6–7 [50]. Differential abundance was calculated with ALDEx2 v 1.38.0 following established pipelines for microbiome data [51,52].

In order to identify whether a gastrovascular/ bell split community partitioning was appropriate for all samples, and identify subcommunity level enterotypes within *Cassiopea* internal and external microbiomes, Dirichlet Multinomial mixture analysis, as described by Holmes and implemented in the DirichletMultinomial package v 1.48 was used for *Cassiopea* – derived samples [53,54]. In short, likelihood models were built based on transformed (log10) abundances for ASVs, then an optimal number of distinct community types was selected based on minimum Laplace values ("mcapply(dmn)").

New models were then trained under known variable constraints (GVC or bell sample identity), and cross-validated ("cvdmngroup()").

In order to identify taxa consistently present across samples of the bell and gastrovascular cavity, core microbiome was calculated for internal and external communities of *Cassiopea*. Core microbiome was computed with the microbiome package v 1.28.0 [55]. For environmental variable interpretation, a redundancy analysis was conducted on center log ration transformed microbial abundances using the microViz package [56].

For identification of *Vibrio* species, the top six ASVs assigned to *Vibrio coralliilyticus* were compared to the NCBI rRNA/ITS database using megablast.

## Ethics statement

This work was performed on lower level invertebrates not subject to Animal Use Protocols or within the purview of Texas A&M University IRB approval. Permitting for these specimens was waived by Florida Fish and Wildlife Conservation Commision.

## Results

### Data description

We analyzed paired microbiome samples associated with 30 medusae from eight sites within the Florida Keys. For each medusa, we analyzed the microbial composition of their gastrovascular cavity ("GVC") and external or bell mucus (herein referred to as "bell"), as well as their substrate ("Sub") or water environment ("WTR"). The four collected paired *C. andromeda* sample sets were excluded from the primary dataset, however they showed no divergence from the trends of the *C. xamachana* simultaneously collected (Supplementary table S1 in S1 File, S fig2 in S2 File).

After filtering, 10.13 and 9.07 million reads were retained ("filterAndTrim") and merged ("mergePairs") respectively (for all sequence quality statistics, see supplementary table S2 in S1 File). There were 8.91 million sequences after chimera removal. After primary phyloseq filtration, the dataset included 6.47 million sequences retained in 34,808 unique ASVs across 76 total samples (30 bell, 30 GVC, 8 water, 8 substrate). Reference sequences for top ASVs are also available in the supplementary material (table S3) in S1 File.

### Distribution of sequences in full dataset

The dataset generated was predominantly bacterial with few archaeal reads (0.7% of sequences were from Archaea lineages). At the phylum level, 57% of sequences were Pseudomonadota (formerly Proteobacteria), 7% were Bacillota (Firmicutes), 13% were Bacteroidota, and 4% were Cyanobacteriota (Fig 2). No other phylum had greater than 3% relative abundance in the dataset.

At the class level, Gammaproteobacteria comprised 41% of all sequences. Alphaproteobacteria, the next most abundant, comprised 17%. Bacteriodia (12%), Bacilli (6%), Cyanobacteriia (4%), and Desulfobacteria (3%), were the other classes with a relative abundance above 2%. No Archaean group was prevalent above a relative abundance of 0.25%, with the three most abundant phyla being Halobacterota (0.25%), Thermoplasmatota (0.21%) and Nanoarchaeota (0.22%).

### Distribution of sequences within gastrovascular cavity samples

Within gastrovascular cavity samples, 75% of all sequences belonged to just three orders, Pseudomonadales (39%), Mycoplasmatales (24%), and Enterobacterales (13%). The majority of Pseudomonadales sequences identified within the GVC were dispersed across very few OTUs, with 66% identified as *Endozoicomonas atrinae* ASV1 and 8% as *E. atrinae* ASV4. This was true of the Mycoplasmatales as well, with 77% of associated gut sequences identified as *Mycoplasma* sp. ASV2 or *Mycoplasma* sp. ASV9 (15%).

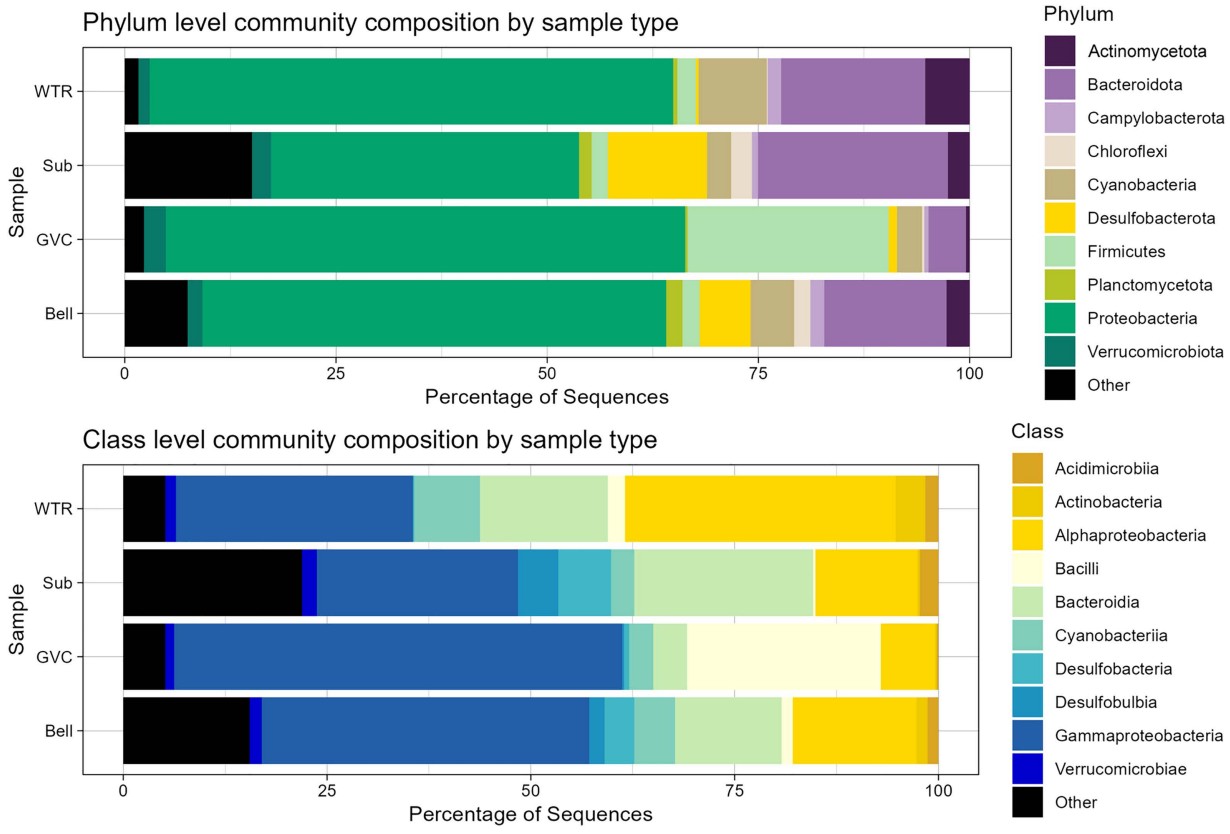

**Fig 2. Top ten phyla and class relative abundances of sequences in samples aggregated by sample type. Gastrovascular cavity and bell samples are n = 30 samples. Water and sediment samples are n = 8.**

## Distribution of sequences within bell mucus samples

At the phylum level, within the bell mucosal swabs, 55% of the sequences were Pseudomonadota, 14.5% Bacteroidota, 5.1% Cyanobacteriota, and 6% Thermodesulfobacteriota (Desulfobacterota).

At the order level, the most abundant groups were Pseudomonadales (23%), Chromatiales (6%), Rhodobacterales (7%), Desulfobacterales (5%), and Flavobacteriales (5%). *Endozoicomonas atrinae* ASV1 was the single most abundant surface mucus ASV at a mean of 12%. The next most abundant were *Synechococcus* CC9902 ASV15 (1.3%), *Endozoicomonas atrinae* ASV4 (1.1%) and Chromatiales *Candidatus Thiobios* ASV6 (0.8%).

## Substrate and water samples

Water samples shared some ASVs with the bell mucus of medusae, the largest contributing ASV was Chromatiales *Candidatus Thiobios* sp. ASV6 (8%). Additional prominent taxa were Rhodobacteraceae *HIMB11* ASV12 (6%), *HIMB11* ASV19 (5%), and SAR11 Clade III ASV 10 (4%). By contrast, substrate samples had no ASVs surpassing a mean of 1% abundance across sites, with the highest two ASVs Sandaracinaceae ASV94 and *Actibacter* ASV105 reaching 0.6% mean abundance across sediment samples (see Table 2; extended table is Table S4 in S1 File). Overlap between bell and sediment samples was greater than between any other sample groups (Upset plot of overlaps in sfig. 3)

## Diversity

Shannon diversity varied by sample type (Kruskal-Wallis: chi-squared = 48.8, df = 3, p-value = 1.412e-10). Shannon diversity indices for the gastrovascular samples varied from substrate samples and bell samples (Kruskal-Wallis Dunn test with Holm correction: GVC-Bell, p-val = 0.00; GVC-Sub, p-val = 0.00; GVC-Water, p-val = 0.07; Bell-Sub, p-val = 0.07; Bell-Water, p-val = 0.06; Water-Sub, p-val = 0.01). Shannon diversity was not explained by location (Kruskal-Wallis: chi-squared = 9.7, df = 7, p-value = 0.21)(Fig 3, full diversity metrics tables in supplement s5 and s6 in S1 File).

Faith's phylogenetic diversity of communities were also distinct from each other (Kruskal-Wallis test, chi-squared = 44.1, df = 3, p-value = 1.4e-09) (Fig 4). GVC communities were distinct from bell and substrate comparison groups, while bells were distinct from water and gastrovascular cavities (Dunn test with holm correction) (Fig 5, s7 in S2 File).

## Beta diversity

Bray-Curtis distance showed differences between medusa and non-medusa samples (pairwise.adonis, GVC vs bell: p = 0.001, bell vs sub: p = 0.008, bell vs water: p = 0.001, GVC vs sub: p = 0.001, GVC vs water: p = 0.001, sub v water p = 0.001). Variability along the primary axis was found in both GVC and bell samples, while variability along the secondary axis was found mainly in GVC samples. Only one pair of gastrovascular cavity sampling sites of Marathon Key (MK) and Key Largo (KL) were significantly different (pairwise Adonis, df = 1, f = 4.63, p = 0.007). The external microbiomes of Marathon Key individuals were significantly different than those of Key West (GB) (pairwise Adonis, df = 1, f = 2.28, p = 0.009), Tavernier (HHP) (pairwise Adonis, df = 1, f = 2.29, p = 0.007), and Bahia Honda Key (pairwise Adonis, df = 1, f = 5.14, p = 0.003) (full adonis comparisons in S8).

## Core microbiome

Core microbiome for the purposes of this study was evaluated based on 70% occurrence of an ASV and a relative abundance of > 0.1% of reads in all 34 gastrovascular cavity samples or external bell samples. This is in line with thresholds for other core microbiome work, however it is important to reiterate that this work was conducted only during one trip, and may be representative of a seasonal community assemblage or one restricted to this specific collection year [57]. The most abundant ASV, *Endozoicomonas* ASV1, was present in 90% of GVC samples at a relative abundance of > 0.1% of reads, ASV2 and ASV4 were present in 80% (Fig 6). Using less stringent criteria (>70% of samples, any abundance),

**Table 2. Top five amplicon sequence variants for each sample type.** Table with ASVs ordered from highest to lowest average proportion in samples of each type displayed as a percentage of composition within that sample type. Includes Silva 138 taxanomic assignment for each ASV.

|  | 1 | 2 | 3 | 4 | 5 |
|---|---|---|---|---|---|
|  | ASV1 *Endozoicomonas atrinae* | ASV15 *Synechococcus* CC9902 | ASV4 *Endozoicomonas atrinae* | ASV6 *Candidatus Thiobios* | ASV10 SAR11 *Clade III* |
| Bell | 11.585% | 1.272% | 1.105% | 0.832% | 0.709% |
|  | ASV94 Sandaracinaceae | ASV105 *Actibacter* | ASV95 Desulfocapsaceae | ASV169 *Robiginitalea* | ASV358 *Clostridiisalibacter* |
| Sub | 0.625% | 0.605% | 0.566% | 0.492% | 0.474% |
|  | ASV6 *Candidatus Thiobios* | ASV12 Rhodobacteraceae *HIMB11* | ASV19 Rhodobacteraceae *HIMB11* | ASV10 SAR11 *Clade III* | ASV18 *Marinobacterium* |
| WTR | 7.521% | 5.839% | 4.725% | 4.497% | 2.711% |
|  | ASV1 *Endozoicomonas atrinae* | ASV2 *Mycoplasma* | ASV9 *Mycoplasma* | ASV4 *Endozoicomonas atrinae* | ASV11 *Vibrio coralliilyticus* |
| GVC | 34.133% | 20.099% | 2.823% | 2.327% | 2.094% |

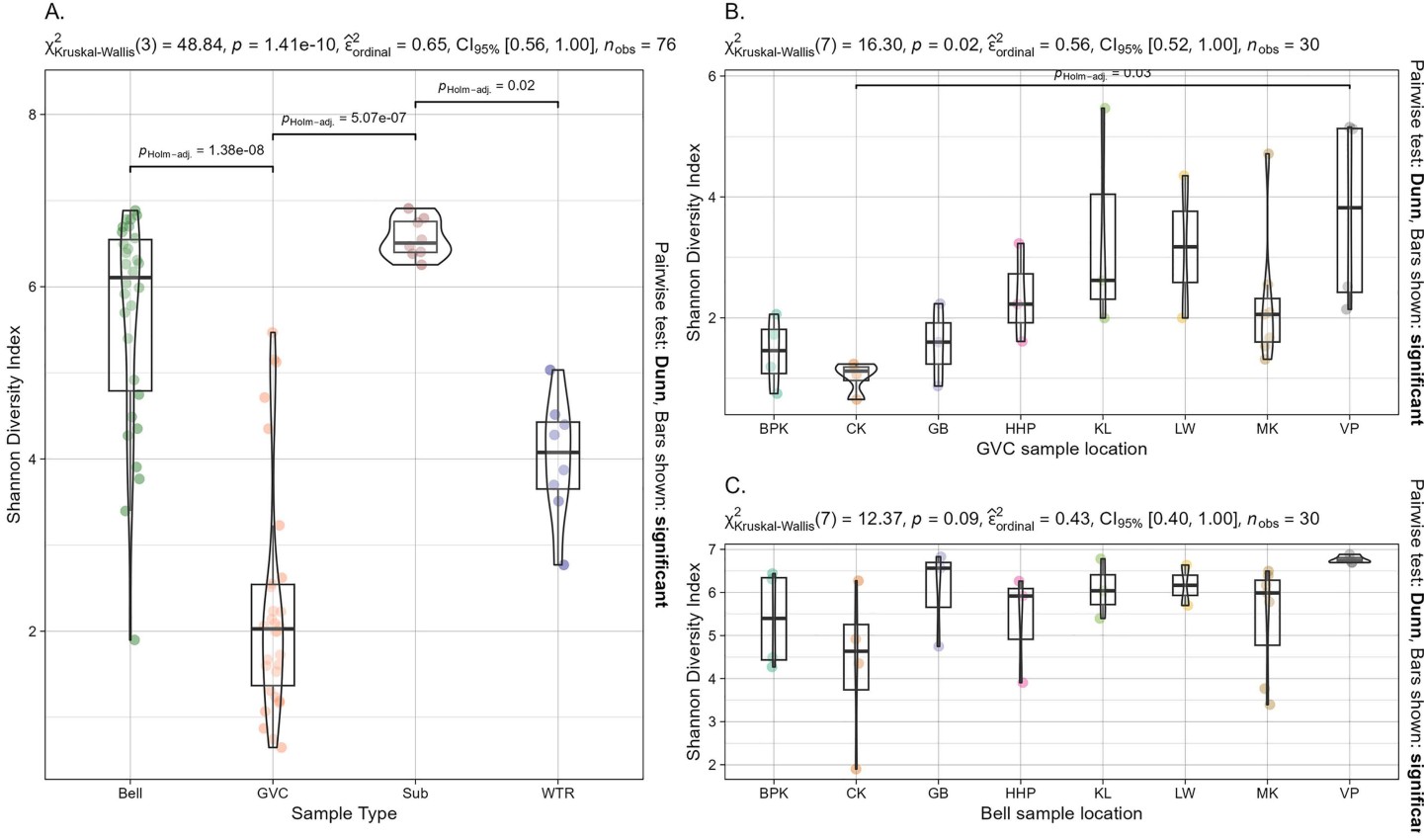

**Fig 3. Shannon diversity by sample type (a) and location (b + c).** Gastrovascular cavity, bell, substrate and water Shannon diversity are shown across both the entire dataset and when segmented by sample type across sites with GVC (b) and bell (c) alpha diversity across sites. All p values of holm-adj Dunn comparison of means tests below a = 0.05 are displayed, with statistical test details displayed above each plot. Means and interquartile ranges are marked with box plots for each sample type.

ASV 11, ASV 132 and ASV 6 become part of the core microbiome. These ASVs are across only three genera- *Endozoicomonas*, *Mycoplasma* and *Vibrio* (full taxonomic details of ASVs can be found in Table S9 in S1 File).

In bell samples, ASV1, ASV104, and ASV86 have 73%, 70% and 70% relative abundance in samples at a 0.1% threshold. ASV6, ASV95, ASV37, ASV105, and ASV135 are present at any level in >70% of bell samples. These ASVs cover the genera *Endozoicomonas*, *Candidatus Thiobios*, *Methyloceanibacter, Ruegeria* and unknown genera from Desulfocapsaceae and Actibacter (Fig S4 in S2 File).

In terms of abundance, internal core microbiome components are not just present, but highly dominant in *Cassiopea*'s gut. Some of these taxa are highly restricted to GVC samples (*Mycoplasma*), while others are prevalent across body surfaces (*Endozoicomonas*) (Fig S4 in S2 File).

The vast majority of ASVs were not differentially abundant between bell and GVC samples. Differential abundance (Aldex2, min effect size absolute value>[±2]) between the gastric and bell surfaces is restricted to the novel *Mycoplasma* group (ASV2), which is completely absent in external samples (Fig 8, Sfig 5 in S2 File). No features were differentially abundant between substrate and bell samples (Sfig 6 in S2 File).

Dirichlet multinomial mixture (DMM) analysis identified specific proportions of many of these core microbiome as valuable for category differentiation. Within animal-associated samples, naïve model probability value was optimal for two communities, one GVC and one bell without subcommunities (k = 2, Laplace = 95583) (Fig 7a). After training on separated

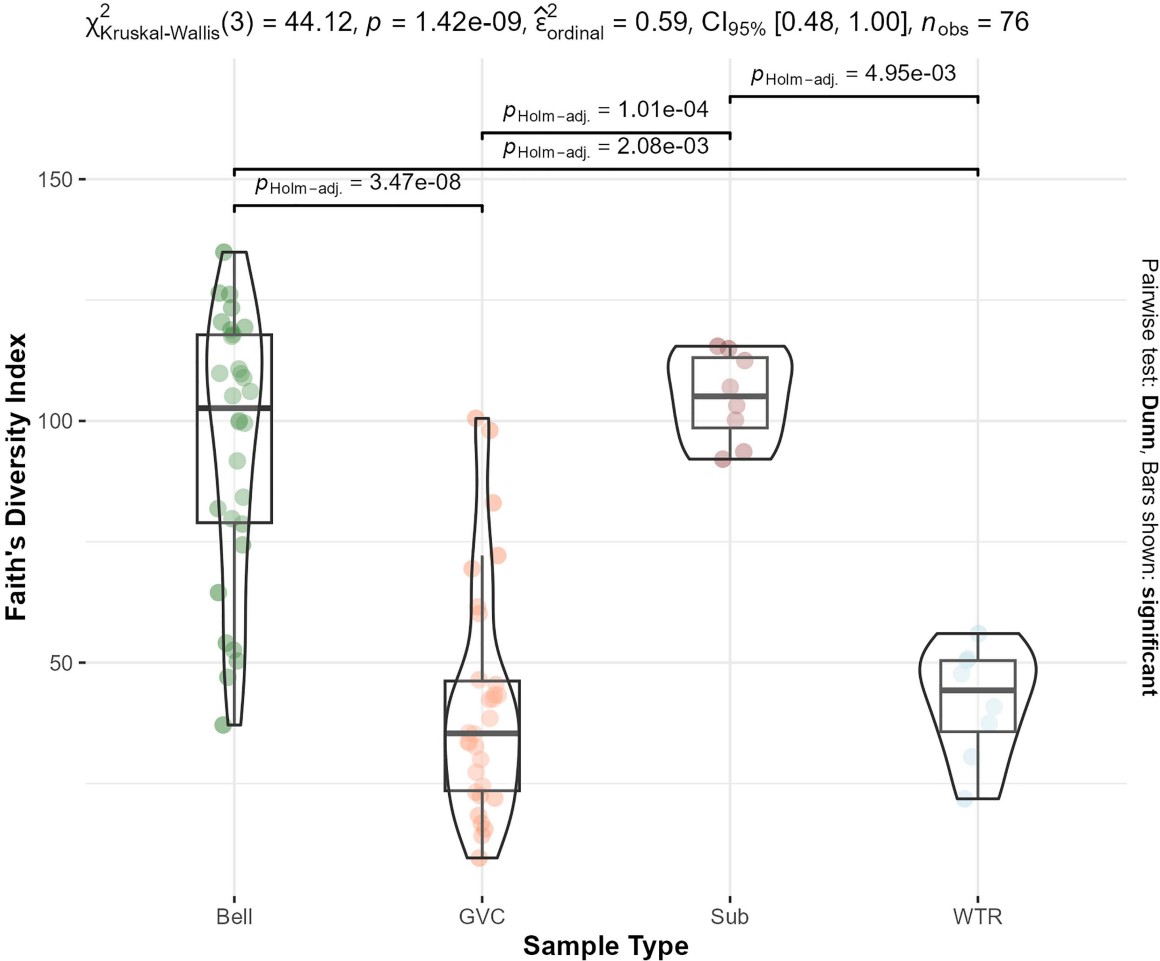

$$\chi^2_{\text{Kruskal-Wallis}}(3) = 44.12, \, p = 1.42\text{e-}09, \, \hat{\varepsilon}^2_{\text{ordinal}} = 0.59, \, CI_{95\%} \, [0.48, 1.00], \, n_{\text{obs}} = 76$$

**Fig 4. Violin plot of Faith's phylogenetic diversity.** Kruskal-Wallis chi-sq (44.12), pvalue (1.4e-9) and pairwise dunn test with holm correction p-values are displayed. Differences between GVC-water and substrate-bell are not significant. Samples are colored by sample type. Means and interquartile ranges are marked with box plots for each sample type.

gastrovascular and bell samples, overall model likelihood was similar and a two community structure was retained (Bell model: k = 1, Laplace = 63910; GVC model: k = 1, Laplace = 33200). ASV1 amount was a core group differentiator, with fewer meaningful ASVs defining the GVC community than the bell community (Fig 7b, 7c). After cross-validation, some bell and gastric samples remained grouped (Fig 7d.).

Mean proportions of *Endozoicomonas*, *Mycoplasma* and *Vibrio* varied across locations. Variation of bell samples primarily occurred with proportion of *Endozoicomonas.* The communities across sites for gastrovascular samples varied in degree of community balance, resolving into primarily Mycoplasmatales-dominated and Pseudomonadales-dominated arrangements, with limited relationships between primary ecological vectors (pH, salinity, temp) and Mycoplasmatales, Vibrionales or Pseudomonadales (Fig 8, Sfig. 7–9 in S2 File).

## Vibrio

ASVs whose closest match within the Silva 138.1 database was to *Vibrio coralliilyticus* were found across *Cassiopea* gastric samples, with abundance of up to 10% of reads within the sample. This far surpasses rates in the surrounding water

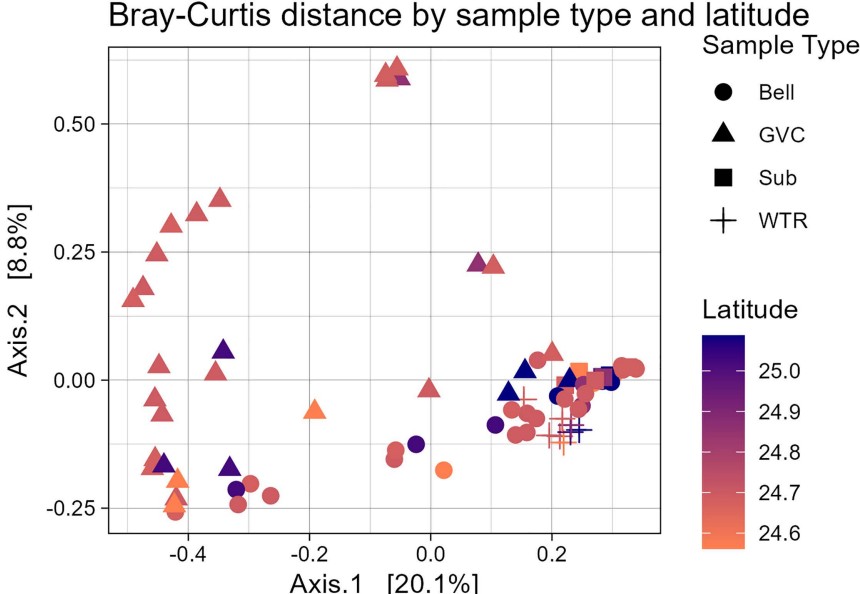

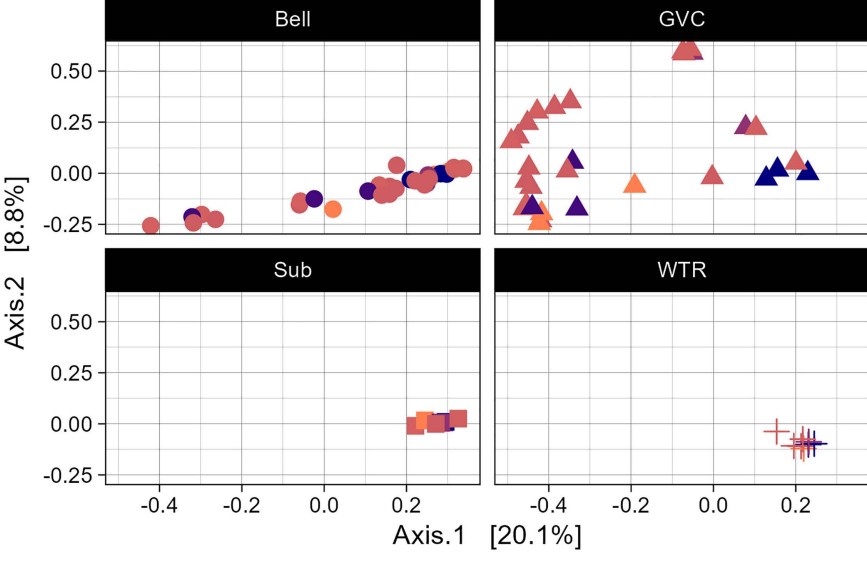

**Fig 5. Principal component analysis (PCA) of Bray-Curtis distance of samples.** PCA is colored by sample collection latitude from lowest (orange) to highest (purple). PCA is presented both together and faceted by sample type. Sample type is designated by shape, circle (Bell), triangle (GVC), square (Substrate) and cross (Water). Bottom plot is presented to demonstrate relative consistency of substrate (Sub) and water (WTR) samples on primary axes 1 and 2.

and substrate samples (ASV11 averages: Bell: 0.05%, WTR: 0.03%, Sub: 0%, GVC: 2.09%) (Fig 9). These ASVs were closely related or identical to *V. tubiashii* and *V. coralliilyticus* within the NCBI rRNA database (Sfig 10 in S2 File).

## Discussion

*Cassiopea* medusae show consistent differences in the microbial communities present on their bell and gastric surfaces. Gastrovascular cavity samples have lower diversity than the sediment, external surfaces, and host a novel *Mycoplasma*

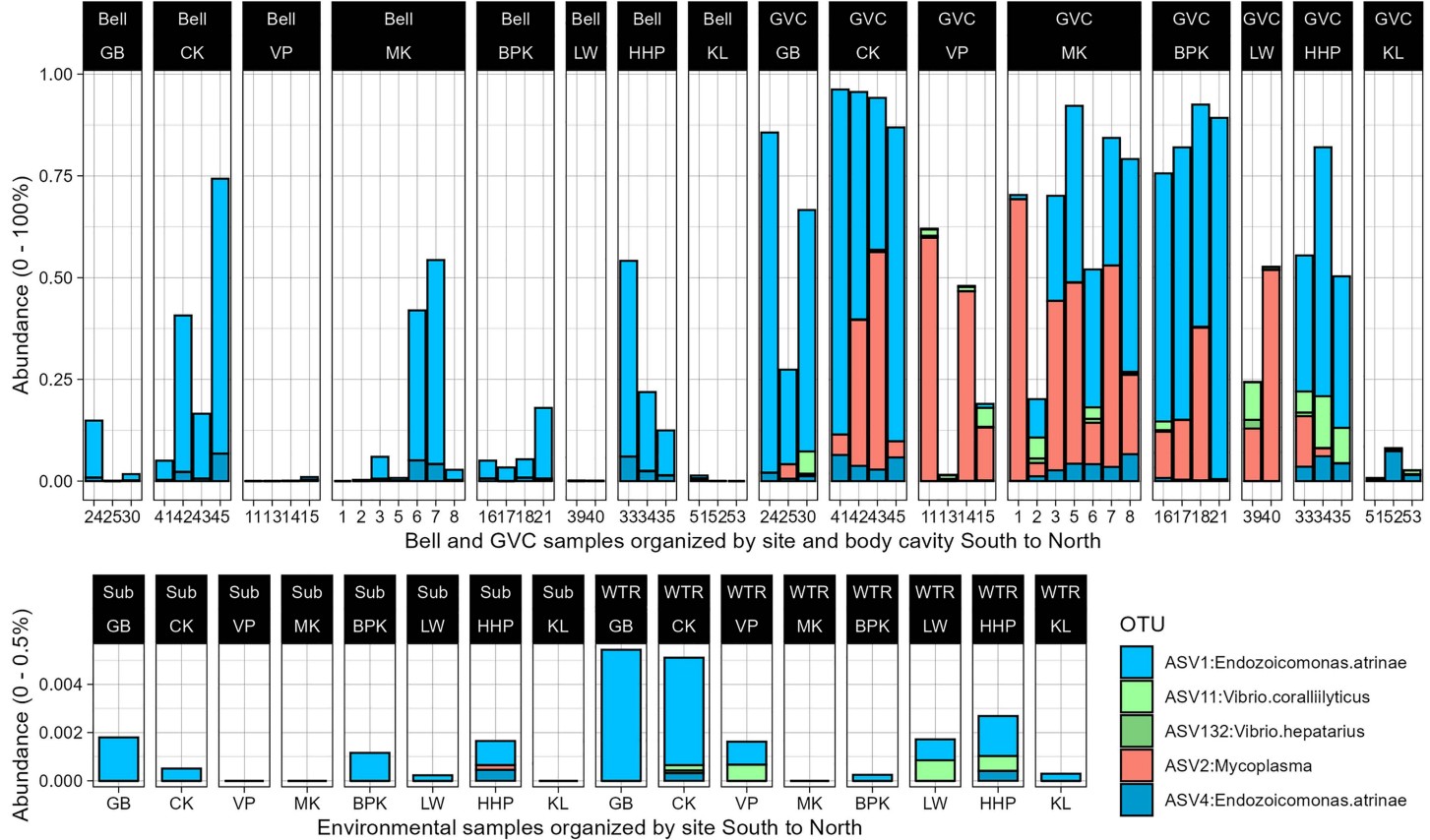

**Fig 6. Abundance of *Cassiopea* core GVC microbiome across dataset.** The top five amplicon sequence variants from the gut core microbiome across all sample types (Bell, GVC, Substrate (Sub) and Water (WTR)) organized from furthest south to furthest north sites. Note the difference in y axis maxima between bell and GVC samples (top) and substrate and water samples (bottom). See Table 1 for site details, the sites are Key West (GB), Cudjoe Key (CK), Big Pine Key (BPK), Bahia Honda Key (VP), Marathon (MK), Lower Matacumbe (LW), Tavernier (HHP), and Key Largo (KL).

lineage. This low diversity is consistent with results for multiple other scyphozoans— including *Cotylorhiza tuberculata* (Macri, 1778) [27]. *Cassiopea* also appears to have some variation in community composition between sites, a feature consistent with results from other species, such as *Aurelia, Mastigias,* and *Tipedalia* [22,26].

## A bifurcated outer and inner microbiome

Across all sites, the gastrovascular cavities of *Cassiopea* medusae showed a core microbiome comprised of taxa previously found in photosymbiotic cnidarians. The external microbial community of *Cassiopea* from the Florida Keys adheres fairly closely to the epibenthic microbial community at each site, with the addition of *Endozoicomonas*. ASV1 *Endozoicomonas* cf. *atrinae* may be a primary mucosal component and extend to the exterior of the body through mucus exchange, as it is less prevalent on the bell surface than internal for most individuals. Alternitavely, ASV1 may be a common mucosal resident with no function for the host.

The abundant *Mycoplasma* and *Vibrio* cf. *coralliilyticus* from the gastric cavity of *Cassiopea* were not found in any external mucus samples and found at far lower abundances in external mucosal samples respectively.

The high diversity of the *Cassiopea* bell microbiome found in this study may be unique to *Cassiopea*. Previous work has reported in the low-diversity mucosa, oral arms and umbrella of a previously studied pelagic rhizostome, *Rhizostoma*

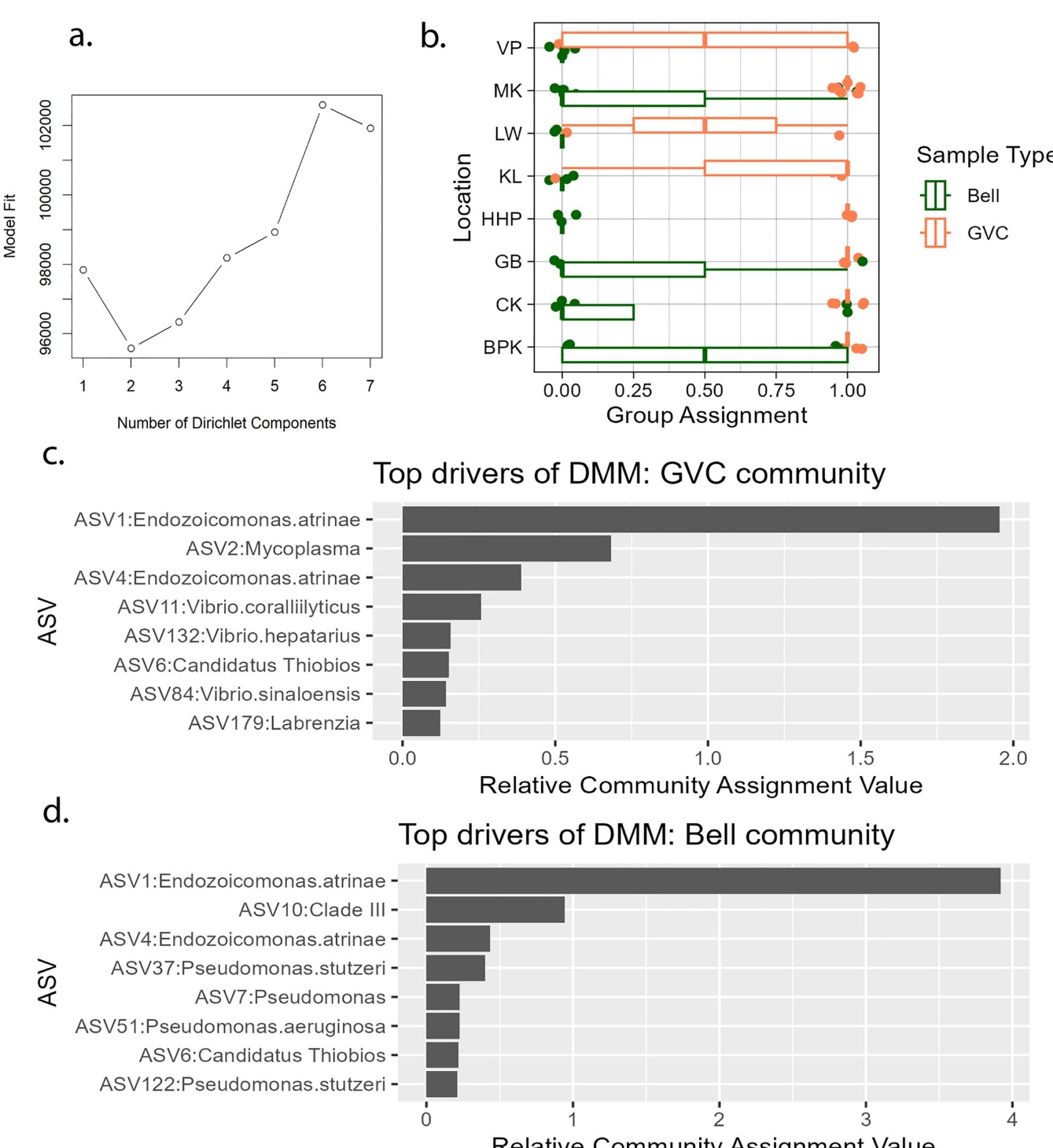

**Fig 7. Dirichlet multinomial modeling model fit (a), community assignments (b), and impactful amplicon sequence variants differentiating the two community types (c+d).** Dirichilet model choice graph (a) displays number of Dirichlet components (distinct assemblage types) tested vs model fit (Laplace value) in *Cassiopea* derived samples for a tested k value of 1-7. Best fit is at 2 components, or k=2. Assignment of samples to one of two communities (b) is shown separated by location and sample type with a "0" value indicative of the bell community type and a "1" value indicative of the GVC community type. Boxplots display means for each sampling group. Points are displayed with a small jitter (0.1) for visibility. ASV list (b) displays the top 0.5% most impactful drivers of body cavity prediction in GVC (c) and bell (d) Dirichlet multinomial models ordered from largest to smallest effect size.

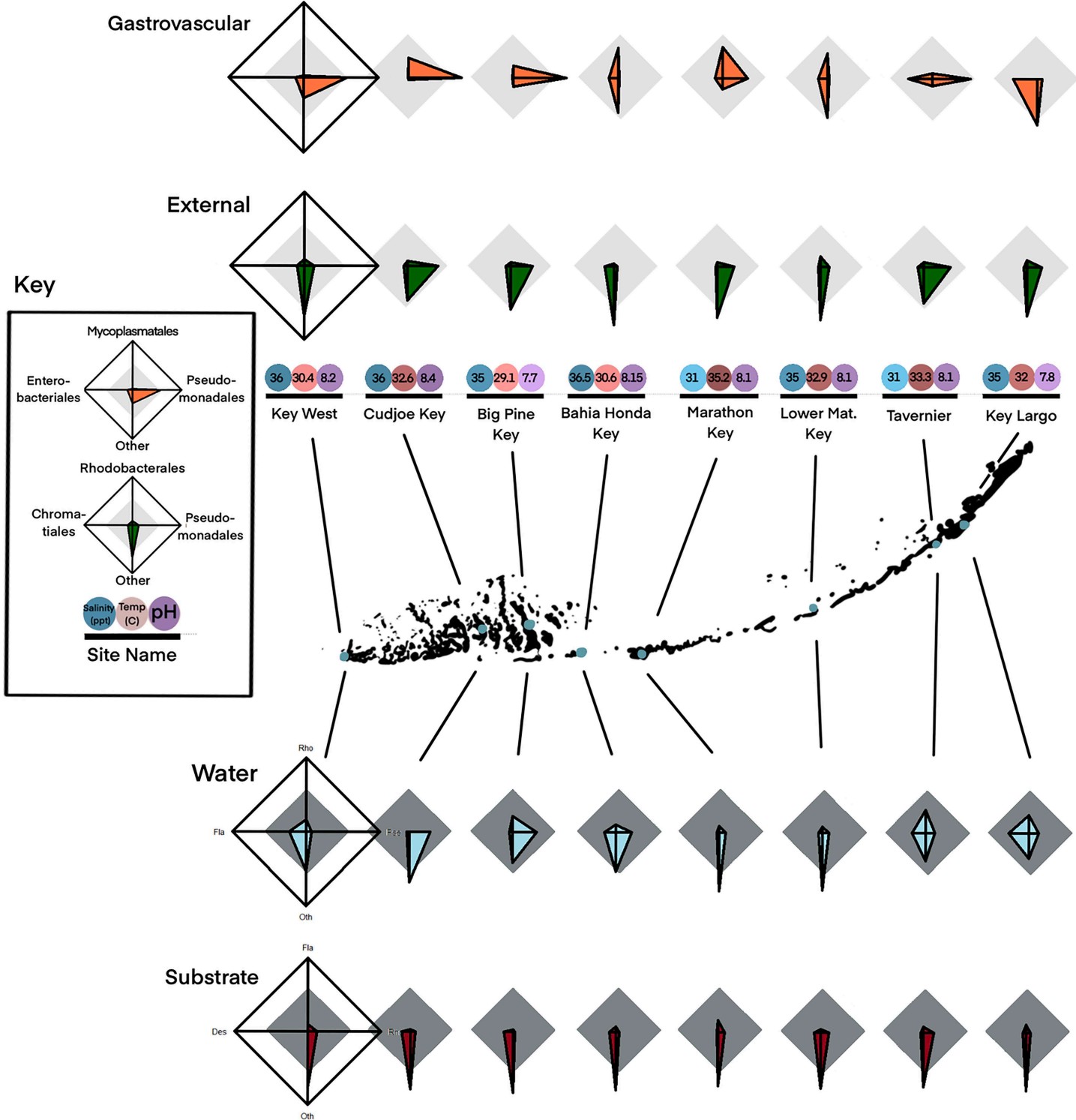

**Fig 8. Star plots of community composition across sites.** Star plots for internal (GVC), external (Bell) mucus swabs, water (WTR) and substrate (Sub) sample averages at each site separated by order (three most prominent orders and "other"). Each point of the star plot ranges from 0% (center) to 100%. **Internal** group, starting from the 3' position and going counterclockwise: Pseudomonadales, Mycoplasmatales, Enterobacteriales and "Other". **External** group points (going counterclockwise): Pseudomonadales, Rhodobacterales, Chromatiales, and "Other". **Water** group points (going

counterclockwise): Pseudomonadales, Rhodobacterales, Flavobacteriales, and "Other". **Substrate** group points (going counterclockwise): Rhodobacterales, Flavobacteriales, Desulfobacterales, and "Other". Each site's measured environmental conditions (salinity, water surface temperature, and pH) are presented below star plots, colored by intensity (low to high represented by pale to dark). Sites are mapped onto their location within the Florida Keys.

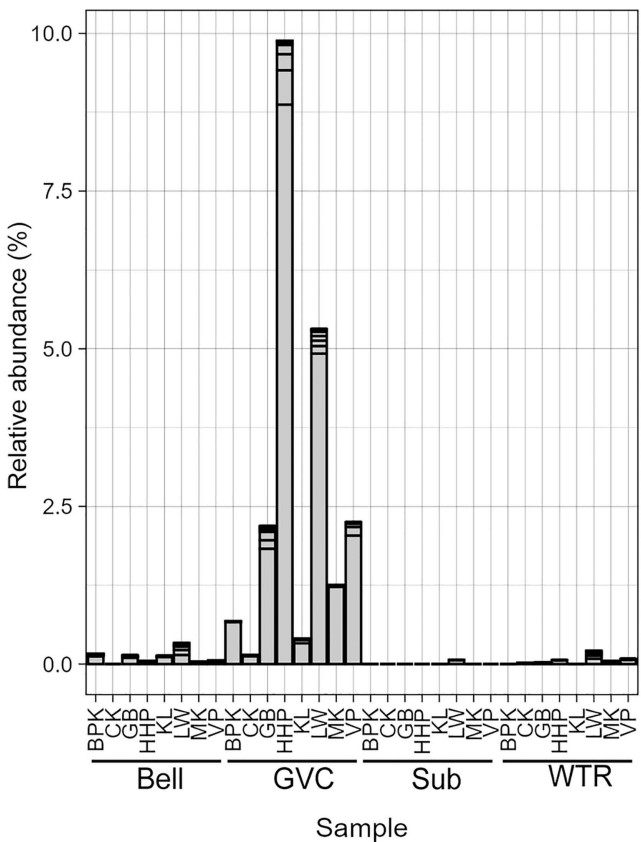

**Fig 9. Average percent of reads (0%−10%) assigned to *V. coralliilyticus* amplicon sequence variants in bell, GVC, substrate and water samples within each site.** Bars are organized by body cavity. Horizontal lines demarcate separations between conspecific ASVs.

*pulmo* Macri, 1778, a likely result of the constant contact of *Cassiopea* with the epibenthos [28]. The limited overlap between diverse bell communities and low diversity gut communities in *Cassiopea* is notable and consistent with the significant differences in body compartment microbiome of other scyphozoans and anthozoans [23,26,58]. While the low diversity in scyphozoan microbiomes may seem distinct from many described stony coral microbial communities, these stony coral microbial profiles are often produced using whole body sampling. Lower diversity gastrovascular cavities with distinct community composition may also be a feature of stony corals [59–61].

Few microbial communities across sites within this study were significantly different from each other, as gastrovascular cavity components remained dominated by three taxa throughout the Keys. This microbiome bears limited resemblance to those sampled in common garden experiments and no resemblance to lab-held lines, demonstrating that "core" here should not be misconstrued as obligate [21,62]. The lack of obligate bacterial taxa in the *Cassiopea* gastrovascular cavity is also consistent with other microbial hosts, notably the anemone *Exaiptasia diaphana* Rapp, 1829, another successful model cnidarian [1]. While *Cassiopea* require their algal symbiont, bacterial partners may be transient, and bacteria may

have far less significance to host health for *Cassiopea* than for specialized obligate hosts (e.g., leafhoppers) [63–66]. Without a better understanding of specific metabolic function of different bacterial associates, microbiome importance cannot be determined. We identified *Endozoicomonas*, *Mycoplasma,* and *Vibrio* as the three most common bacterial genera found within the gastrovascular cavity of *Cassiopea*. These are all commonly associated with coral microbial communities [8,9,67]. Below, we provide information on these common bacterial components of cnidarian holobionts.

### *Endozoicomonas* functions in microbiomes

The large proportion of *Endozoicomonas* in internal and external samples fits within a broader framework of both *Endozoicomonas* and known cnidarian-associated microbial communities; *Endozoicomonas* strains are present in a variety of invertebrates, including many corals, sponges, sea anemones, scyphozoans and cubozoans [9,67–70]. Some hosts house multiple distinct *Endozoicomonas* genotypes, suggesting that different *Endozoicomonas* clades may provide different metabolic services to a host (e.g., Vitamin B12 metabolism) [71]. *Endozoicomonas* is more commonly identified in healthy photosymbiotic scleractinian corals than bleached and diseased ones [72]. For these corals, it may play a role in a variety of fundamental *Symbiodinium*–host interactions, including nutrient transport and symbiont cell breakdown [71]. Notably, *Endozoicomonas* is not a consistent marker of microbiome health, its disappearance and reappearance without apparent host health declines has been identified in octocorals [2,70]. In some coral, members of *Endozoicomonas* are found in close proximity to *Symbiodinium,* it is unknown whether this feature is shared with *Cassiopea* [71]. As few cnidarians previously studied have gastric cavities large enough to be readily sampled, the finding of *Endozoicomonas* across cavities suggests that this group is not specialized in surface defense.

Within scyphozoa, *Endozoicomonas* is abundant in the microbial community of the rhizostome scyphozoan *Mastigias* [22]. While *Mastigias*, like *Cassiopea*, host zooxanthellae from the family Symbiodiniaceae, not all symbiotic rhizostomes harbor the microbe, and not all taxa with the microbe are symbiotic [22,27]. *Endozoicomonas* may be common across medusae, but the difficulty of culturing some strains may have excluded them from the results of past culture-based analyses [70,71]. Greater clarity is needed on the function of *Endozoicomonas* for these medusae and other taxa.

### Mycoplasmataceae: a poorly understood cnidarian microbiome resident

*Mycoplasma*, like *Endozoicomonas*, is present in studied coral and scyphozoan microbiomes [9,24,25,27,73]. While intracellular parasitism is common for members of Mycoplasmataceae, those associated with cnidarians may be located extracellularly and exist as opportunistic commensals [2,74]. *Mycoplasma* sp. is noted as a potential endosymbiont in lab-raised *Aurelia* polyps, while adults from many jellyfish populations are known to have them [24,26,29]. The high divergence between this *Mycoplasma* and others of the genus (closest sequence identity on NCBI at 89%: Accn. No. NR_115937) suggests that this may be a hitherto unknown group. Whether this *Mycoplasma* represents an endosymbiont, a result of prey capture, or a parasite, its exclusivity to the gastrovascular tract in *Cassiopea* sampled within the Keys is notable, as other studies have not demonstrated gastrovascular cavity localization [26,29].

### Vibrionaceae

*Vibrio*naceae is both a core part of coral microbiomes [9,75] and known to facilitate metamorphosis within *Cassiopea* [76,77]*,* however, it is a highly diverse family. While *Vibrio* spp. are found in gut communities of presumed healthy scyphozoans [24,25] they can also be associated with senescence and disease [78,79]. For *Cassiopea* specifically, an increase in *Vibrio* has previously been associated with an aposymbiotic or stressed state [62]. Given the variation in Vibrionaceae species and strains, family-wide statements on the beneficial or harmful nature of Vibrionaceae within *Cassiopea* gastrovascular cavities should not be made.

### Vibrio coralliilyticus

In this dataset we found ample signal of a 16S V3-V4 with 100% sequence match to multiple *Vibrio* strains, including *V. coralliilyticus* and the closely related oyster pathogen, *V. tubiashii*. As *Vibrio* species are highly varied, the sequence match does not mean assured pathogenicity towards any taxa [80–82]. The apparent health of the medusae from whom these samples were collected suggests either a lack of impact of this specific group on the host or a host-microbe facultative interaction between this *Vibrio* and *Cassiopea.* While there is data on *Cassiopea* microbiome antimicrobial isolates and *Cassiopea* mortality when faced with *Serratia marcescens* infection, we lack adequate details on the *Vibrio – Cassiopea* system to determine whether this represents a low abundance widespread infection or a mutualism [83,84].

### Drivers of external microbiome

While the external microbiome was not distinct from the sediment around it, it is worth discussing the major groups that are more common externally than internally. While the core bell microbiome represents a substrate-like community, the bell core microbiome includes representatives from families and genera with a variety of potential metabolic capacities. The most plentiful of these ASVs was *Candidatus Thiobios*, known as a sulfur-oxidizing epibiont of the *Zoothamnium niveum* [85,86]. As this bacteria can be found in the Florida Keys, their appearance on *Cassiopea* may be circumstantial, but is consistent across medusae [85]. Isolates from *Methyloceanibacter* have been demonstrated to be methane oxidizers from marine environments [87]. These may be sediment associated or may recruit to the host to utilize algal products. The potential sulfate reducers within Desulfocapaceae may similarly recruit to the bell and its access to sediment pore waters [88]. While *Pseudomonas* cf. *stuzeri* was not as high abundance as the other metabolizers listed above, it may have a role in denitrification or coral health for some species [89–91]. While all of these groups were low abundance within host mucosa, they may still be important community members, as the importance of rare microbes is an ongoing topic of study within cnidarian microbiomes [73,92,93].

In addition to microbes with known metabolic functions that align with the *Cassiopea* holobiont, the genus *Ruegeria* includes many marine species and has been identified in the microbiomes of photosymbiotic and nonphotosymbiotic cnidarians alike [9,78,94]. As Dirichlet modeling for bell microbiomes gives similar weight to many bacteria, the relative impact of all of these bacterial epibionts on the host is unclear.

### Laboratory cultures

*Cassiopea* are primarily used for in-lab studies on symbiosis [12,14,95]. Some of these use wild-caught individuals, and others use lab-bred cultures, many of which have been held for years. The lab-raised *Cassiopea* used in Rothig et al. 2021 appear to have a completely different microbiome than the wild individuals caught here [21]. This potential discrepancy should be considered when using wild-caught or captive-bred individuals, as captive-bred symbiosis and tolerance levels may differ from medusae in the wild. These discrepancies in microbiome are found in other model cnidarians, such as *Exaiptasia* [1]. The information on wild *Cassiopea* microbial communities from the Florida Keys presented here may allow researchers working with lab-raised individuals to better contextualize the taxa that they find in cultured medusae.

### Conclusion

The microbiome of *Cassiopea* within the Florida Keys demonstrates strong body compartment discrimination and includes a central group of ASVs overrepresented in the gut and bell mucus. All of these families, Pseudomonadales, Vibrionaceae, and Mycoplasmataceae, are found in other medusae and anthozoans. Despite this overrepresentation of some microbial taxa within the gut, the *Cassiopea* bell is highly diverse, and overlaps with the communities of the water and substrate in and on which it resides. The consistency of *Cassiopea* microbial communities across the Florida Keys demonstrates that specific taxa are strongly associated with *Cassiopea* and that this can extend across many sites with

diverse physical conditions. This work provides a greater understanding of the wild microbiome of the common lab organism *Cassiopea*.

## Supporting information

**S1 File. Supplementary Tables 1–10.** The set of all supplementary tables for this manuscript. Supplementary Table S1 is the roster of collected medusae. Supplementary Table S2 is the read processing tracking table. Supplementary Table S3 are the representative sequences of all ASVs from the dataset. Supplementary Table S4 is all ASVs orderd by abundance in each sample type. Supplementary Table S5 is alpha diversity of all samples. Supplementary Table S6 is the alpha diversity of all sample types with statistical groupings. Supplementary Table S7 is Faiths diversity by sample. Supplementary Table S8 is the list of statistical comparisons for Adonis2 groupings. Supplementary Table S9 is the taxonomy of top 1000 ASVs in dataset. Supplementary Table S10 are the top *Vibrio* cf. *coralliilyticus* NCBI hits.
(XLSX)

**S2 File. Supplementary figures 1–9 and supplementary description A1.** Sfig1) Rarefaction curves of all samples. Sfig2) Principal component analysis of Bray-Curtis distance between microbiomes of *Cassiopea* species. Sfig 3) Upset plot of overlap between taxa found in bell, GVC, water and substrate samples. Sfig4) Heatmap of core microbiome prevalence in GVC and bell. Sfig5) Aldex2 plot of effect size of all ASVs between bell and GVC samples. Sfig6) Aldex2 plot of effect size of all ASVs between bell and substrate samples. Sfig7) Correlation plot (corrplot) of collected environmental and medusa factors across samples. Sfig8) Redundancy analysis plot of GVC. Sfig9) Redundancy analysis plot of bell samples [96,97].
(DOCX)

## Acknowledgments

The present work was part of KMM's PhD dissertation. Thank you to Dr. David Retchless for input on figure design. We thank the Florida Fish and Wildlife Conservation Commission for their help and guidance.

## Author contributions

**Conceptualization:** Kaden Muffett.

**Formal analysis:** Kaden Muffett.

**Funding acquisition:** Kaden Muffett, Maria Pia Miglietta.

**Investigation:** Kaden Muffett.

**Methodology:** Kaden Muffett, Jessica Labonté.

**Supervision:** Jessica Labonté.

**Visualization:** Kaden Muffett.

**Writing – original draft:** Kaden Muffett.

**Writing – review & editing:** Jessica Labonté, Maria Pia Miglietta.

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
