## [Decision Letter · Decision Letter 0]

28 Oct 2024

Dear Dr. Muffett,

We look forward to receiving your revised manuscript.

Kind regards,

Stefano Piraino

Academic Editor

PLOS ONE

Journal Requirements:

1. When submitting your revision, we need you to address these additional requirements. Please ensure that your manuscript meets PLOS ONE's style requirements, including those for file naming. The PLOS ONE style templates can be found at https://journals.plos.org/plosone/s/file?id=wjVg/PLOSOne_formatting_sample_main_body.pdf and https://journals.plos.org/plosone/s/file?id=ba62/PLOSOne_formatting_sample_title_authors_affiliations.pdf 2. Thank you for stating the following financial disclosure: "This work was funded by Texas Sea Grant Award #NA18OAR4170088 and Texas A&M University Galveston" Please state what role the funders took in the study.  If the funders had no role, please state: "The funders had no role in study design, data collection and analysis, decision to publish, or preparation of the manuscript." If this statement is not correct you must amend it as needed. Please include this amended Role of Funder statement in your cover letter; we will change the online submission form on your behalf. 3. Please note that in order to use the direct billing option the corresponding author must be affiliated with the chosen institute. Please either amend your manuscript to change the affiliation or corresponding author, or email us at plosone@plos.org with a request to remove this option. 4. Please include your full ethics statement in the ‘Methods’ section of your manuscript file. In your statement, please include the full name of the IRB or ethics committee who approved or waived your study, as well as whether or not you obtained informed written or verbal consent. If consent was waived for your study, please include this information in your statement as well. 5. Please upload a new copy of Figure 3 and Sfig1 as the detail is not clear. Please follow the link for more information: https://blogs.plos.org/plos/2019/06/looking-good-tips-for-creating-your-plos-figures-graphics/"" https://blogs.plos.org/plos/2019/06/looking-good-tips-for-creating-your-plos-figures-graphics/""

Additional Editor Comments:

Thank you for submitting your manuscript to PONE. I must As the reviewers recognised, your manuscript can be of interest to the PONE scientific community. However, both raised concerns that prevent me from recommending the publication of your manuscript in its present form. Major issues deal with limited information on methodology (e.g. used statistics, sample size, preservatives), and the design of figures. One of the reviewers also pointed out an overstatement of your results (also from expectations from the current title of your manuscript). As you will see, the number of required improvements will require substantial revision. However, both reviewers provided a detailed review with questions and remarks, which I believe can help you and your coauthors to significantly improve your manuscript. I am confident you will be able to address all points and resubmit your revised manuscript as soon as possible. Best wishes

Stefano Piraino

Reviewers' comments:

Reviewer's Responses to Questions

**Comments to the Author**

1. Is the manuscript technically sound, and do the data support the conclusions?

Reviewer #1: Yes

Reviewer #2: Partly

2. Has the statistical analysis been performed appropriately and rigorously?

Reviewer #1: Yes

Reviewer #2: I Don't Know

3. Have the authors made all data underlying the findings in their manuscript fully available?

Reviewer #1: Yes

Reviewer #2: No

4. Is the manuscript presented in an intelligible fashion and written in standard English?

Reviewer #1: Yes

Reviewer #2: Yes

Reviewer #1: Review of manuscript PONE D-24-22287

Comments:

Abstract – needs to mention the consistent results obtained from the sampling sites.

Results – four medusae were identified as C. andromeda (lines 165-167); it is not clear from the start if these medusae were included in the analysis. Please clarify.

I would suggest calling the “benthic” samples, sediment samples, to avoid confusion.

Figure 2 needs a title. Also, please specify if the pictured abundances are relative or absolute.

Figure 3 refers to the 20 most common orders in each sample. I would suggest removing “diversity” from its title.

The methods need to clarify if the same medusae were analyzed for external and internal samples.

I think the “Dirichlet multinomial analysis” needs a cite/reference.

Results: In line 227 the authors give the statement that clustering difference was most distinct between GVC and environmental samples. But how can you tell which difference is bigger, when internal vs external communities were as different as GVC from sediments and water samples (all with P-value <0.001). Please revise.

Results: In the “Core microbiome” (of the GVC) section, it is difficult to understand if the microbial community of the medusae incorporate the same members when the analysis is done at two different taxonomic levels (Order and Family). However, as I understand it, a core microbiome names its members at least at the genus level. I would suggest renaming this section.

The same argument goes for the first section of the discussion (Line 276).

Discussion: In sections 4.2 to 4.4, discussion pertains only to the bacterial groups identified in the GVC, which needs to be clear. Please revise lines 308-309.

In the discussion there was no mention of results obtained from the external community.

I find paragraph 359-364 unnecessary.

Specific comments:

Line 33: “These communities” please specify you are referring to “microbial communities”

Line 38: replace “clade” with “Family”

Line 47: the statement about the tolerance of Cassiopea to intense temperature fluctuations needs a reference

Line 63: I do not understand the purpose of the statement: “Pathogenesis is often a focus…”

Line 150-151: Why were you not able to use the Siva 138 database?

Line 165: “benthic OR water” should not be “water AND sediment”?

Line 216: Please revise “Family richness was significantly lower within medusae”

Line 227: after referring to your results, it would be good to add a figure number

Lines 280-282: this statement has no support. It could work the other way, going from the exterior but remaining an important component in the internal cavity. Please revise.

Lines 299-300: I do not understand why you mention this statement, since in your results no identification of “obligate” taxa was made for the GVC. Please revise.

Line 314: a space is needed before the parenthesis

Lines 317-319: is not clear what point are you trying to make.

Line 326: use “Family” rather than “clade”

Line 327: orphan phrase? “…appear to harbor the microbiome, …”

Line 381: I suggest changing “lacks a conservatively defined core microbiome” for “in our samples, no core microbiome could be demonstrated”

Reviewer #2: The authors study the microbiome composition of two upside-down jellyfish species in the Florida Keys using 16S rRNA barcoding. The paper describes the main bacterial taxa in different locations and describes the microbiome composition focusing on an understudied group. They briefly compare these findings with the existing literature for other anthozoan groups.

I found the study interesting, but the results seem overstated, and some important information is missing. For example, the title does not accurately reflect the study's findings. It should better represent the study’s scope and avoid claiming general patterns that the data does not appear to support. My major concern is the lack of explanation in the statistical analyses. I think improving this part will help to better understand the analyses and evaluate the content of the manuscript.

Methods

Overall, there is a lack of information on sample sizes throughout the study. This omission makes it difficult to assess the balance of the study design or to understand how many samples are being compared in each analysis. Sample size details should be included in the figures, methods, and results sections.

The authors use two different preservatives—ethanol and DESS—that appear to be combined without distinction. I recommend including a statement in the methods on how this might affect microbiome composition. If the authors tested for potential effects, the results should be included, or they should reference studies clarifying whether these preservatives could influence the results.

Table S1: Please provide descriptions for abbreviations used to aid comprehension.

Table 1: Please add the sample sizes per site.

L129-135: If PacBio sequences were not used in the study, I recommend removing this from the methods section.

L157: Please describe the statistical analyses clearly enough to allow reproducibility. This section lacks essential details.

Figure legends need explanations of the content. They are currently limited to single-line titles, which restricts comprehension. I recommend revising the figure design, e.g. Figure 1.

Results

L226: "significantly different".

Figure 4: The stress level is high. How balanced is this comparison? Although the figure appears to show a comparison analysis, the text refers to a pairwise comparison. Could the authors clarify the type of analysis used here?

The results section presents many analyses and tests that were not introduced in the methods section. Additionally, the results refer to supplementary tables without legends, making it difficult to assess the accuracy of these tests. I recommend a more detailed explanation of the analyses conducted.

Figure 2: The y-axis label is missing.

**Do you want your identity to be public for this peer review?** For information about this choice, including consent withdrawal, please see our Privacy Policy

Reviewer #1: No

Reviewer #2: No

---

## [Author Response · Author response to Decision Letter 1]

28 Jan 2025

Response to editor and reviewers

Dear editor and reviewers,

After careful review of your suggestions and criticisms, the methodological and statistical analyses in this manuscript have been revisited, redone, and in the process greatly improved. Many of your broad concerns in these areas are addressed with a significantly expanded methods section.

Sincerely,

KMM

Response to editor:

Thank you for your considered feedback.

In addition to the improved manuscript, please note these specific changes:

1. Inclusion of the funder role statement along with the manuscript

2. Corrected Funding statement in cover letter

3. Fixed formatting in manuscript

4. Added ethics statement

Comments to reviewer 1:

Thank you for your careful feedback. After substantial revision to the backbone of this work, we hope you will find all of your comments duly taken into consideration. Please see specific corrections in addition to the broader overhaul in the response below.

1. Abstract needs to mention consistent results from across sampling sites.

a. Thank you for your feedback. This has been corrected.

2. Clarify inclusion of C. andromeda in analysis.

a. In the text (ln 146) it now clarifies that C. xamachana only are used for downstream analyses with the exception explicitly comparing the two mitotypes (ln 298).

3. Standardize “benthic” sample name

a. Thank you for your suggestion. All substrate samples are now referred to as substrate as opposed to benthic.

4. Figure 2

a. Figure two has been replaced with an improved version that includes your suggestions.

5. Figure 3

a. Figure 3 has been replaced and now is a chart of Shannon and observed diversity

6. Citation for Dirichlet multinomial analysis

a. The lack of reference to Holme’s original work has been rectified.

7. Line 227: claim about clustering distinction

a. Thank you. This claim has been removed and has been replaced with Faith’s D and more appropriate accompanying statistical tests.

8. Core microbiome information incomplete

a. Thank you for the feedback. This section was indeed incomplete and confusing. It has been replaced with analyses from the “microbiome” r package that are more appropriate.

9. Lack discussion of external microbiome

a. Thank you for this comment. There is now a section discussing this within the paper.

10. Seasonality and size comments unnecessary

a. Thank you for your feedback, this section has been removed.

Specific comments:

• Line 33: “These communities” please specify you are referring to “microbial communities”

o This issue has been noted and corrected.

• Line 38: replace “clade” with “Family”

o This issue has been noted and corrected.

• Line 47: the statement about the tolerance of Cassiopea to intense temperature fluctuations needs a reference

o This issue has been noted and corrected.

• Line 63: I do not understand the purpose of the statement: “Pathogenesis is often a focus…”

o This statement has been clarified and corrected.

• Line 150-151: Why were you not able to use the Siva 138 database?

o This work now uses the most up-to-date silva database.

• Line 165: “benthic OR water” should not be “water AND sediment”?

o This issue has been noted and corrected.

• Line 216: Please revise “Family richness was significantly lower within medusae”

o This line has been corrected.

• Line 227: after referring to your results, it would be good to add a figure number

o Figure numbers are now added in correct positions.

• Lines 280-282: this statement has no support. It could work the other way, going from the exterior but remaining an important component in the internal cavity. Please revise.

o This error has been noted and replaced.

• Lines 299-300: I do not understand why you mention this statement, since in your results no identification of “obligate” taxa was made for the GVC. Please revise.

o This statement has been noted and revised.

• Line 314: a space is needed before the parenthesis

o This issue has been corrected.

• Lines 317-319: is not clear what point are you trying to make.

o Thank you for the feedback, this statement has been removed and replaced with one of greater clarity.

• Line 326: use “Family” rather than “clade”

o Corrected.

• Line 327: orphan phrase? “…appear to harbor the microbiome, …”

o Thank you for the feedback, this statement has been clarified and reworded.

• Line 381: I suggest changing “lacks a conservatively defined core microbiome” for “in our samples, no core microbiome could be demonstrated”

o This statement has been removed and replaced after reanalysis.

Comments to reviewer 2:

The authors thank reviewer 2 for their supportive feedback. The poor explanation of statistical tests and underanalysis of the dataset at hand presented key failings that have been carefully corrected in the updated version. We hope that these revisions make this work more meaningful and improve readability.

Reviewer #2: The authors study the microbiome composition of two upside-down jellyfish species in the Florida Keys using 16S rRNA barcoding. The paper describes the main bacterial taxa in different locations and describes the microbiome composition focusing on an understudied group. They briefly compare these findings with the existing literature for other anthozoan groups.

I found the study interesting, but the results seem overstated, and some important information is missing. For example, the title does not accurately reflect the study's findings. It should better represent the study’s scope and avoid claiming general patterns that the data does not appear to support. My major concern is the lack of explanation in the statistical analyses. I think improving this part will help to better understand the analyses and evaluate the content of the manuscript.

Responses to specific comments:

• Overall, there is a lack of information on sample sizes throughout the study. This omission makes it difficult to assess the balance of the study design or to understand how many samples are being compared in each analysis. Sample size details should be included in the figures, methods, and results sections.

o Thank you for your comment. The sample size information is now added to the first table and where relevant in other sections.

• The authors use two different preservatives—ethanol and DESS—that appear to be combined without distinction. I recommend including a statement in the methods on how this might affect microbiome composition. If the authors tested for potential effects, the results should be included, or they should reference studies clarifying whether these preservatives could influence the results.

o Ethanol was not used for any downstream analyses, a statement that has now been added to the methods section to clarify.

• Table S1: Please provide descriptions for abbreviations used to aid comprehension.

o Clarification has now been added to the supplementary information.

• Table 1: Please add the sample sizes per site.

o Corrected

• L129-135: If PacBio sequences were not used in the study, I recommend removing this from the methods section.

o Corrected.

• L157: Please describe the statistical analyses clearly enough to allow reproducibility. This section lacks essential details.

o Corrected.

• Figure legends need explanations of the content. They are currently limited to single-line titles, which restricts comprehension. I recommend revising the figure design, e.g. Figure 1.

o Corrected

• L226: "significantly different".

o This statement has been replaced and clarified.

• Figure 4: The stress level is high. How balanced is this comparison? Although the figure appears to show a comparison analysis, the text refers to a pairwise comparison. Could the authors clarify the type of analysis used here?

o Thank you for your comment. This analysis has been replaced with a more appropriate MDS of Euclidean distance from the corrected data.

• The results section presents many analyses and tests that were not introduced in the methods section. Additionally, the results refer to supplementary tables without legends, making it difficult to assess the accuracy of these tests. I recommend a more detailed explanation of the analyses conducted.

o This consistent issue has been corrected and all analyses are now listed in the revised version of the manuscript.

• Figure 2: The y-axis label is missing.

o This figure has been replaced with a more appropriate figure.

---

## [Decision Letter · Decision Letter 1]

15 Apr 2025

Dear Dr. Muffett,

Thank you for submitting your manuscript to PLOS ONE. After careful consideration, we feel that it has merit but does not fully meet PLOS ONE’s publication criteria as it currently stands. Therefore, we invite you to submit a revised version of the manuscript that addresses the points raised during the review process.

We look forward to receiving your revised manuscript.

Kind regards,

Sergio N. Stampar, Dr.

Academic Editor

PLOS ONE

Journal Requirements:

Additional Editor Comments:

Based on the reviewers' opinions, I recommend a minor revision so that the manuscript can be accepted after these modifications.

Reviewers' comments:

Reviewer's Responses to Questions

**Comments to the Author**

Reviewer #1: All comments have been addressed

Reviewer #3: (No Response)

Reviewer #4: (No Response)

2. Is the manuscript technically sound, and do the data support the conclusions?

Reviewer #1: Yes

Reviewer #3: Yes

Reviewer #4: Partly

3. Has the statistical analysis been performed appropriately and rigorously?

Reviewer #1: Yes

Reviewer #3: Yes

Reviewer #4: Yes

4. Have the authors made all data underlying the findings in their manuscript fully available?

Reviewer #1: Yes

Reviewer #3: Yes

Reviewer #4: Yes

5. Is the manuscript presented in an intelligible fashion and written in standard English?

Reviewer #1: Yes

Reviewer #3: Yes

Reviewer #4: No

Reviewer #1: Authors have addressed previous comments. However, I have minor issues that need attention. To make this easier, I have attached a pdf file signaling such issues.

Briefly, since the samples preserved in alcohol did not give good results, these should be removed from the manuscript.

Table 1 needs work with headings, column spaces, and some definitions. The GVC term should be introduced in Methods.

The authors collected information on physical parameters, but such information was not relevant in their analyses, correct?

Finally, in the Results section referring to Core Microbiome, the authors need to stress that the sampling was done in one month within a single year.

Reviewer #3: This is a very interesting study on the internal and external microbiome of Cassiopea xamachana from Florida Keys.

I see that this work has already been reviewed and duly revised by the authors, therefore I only have some minor suggestions and comments, and recommend its publication.

- Title: because both internal and external microbiomes were investigated, the words “gastric microbiome” limits the scope of the study, which is broader. My suggestion is to alter it.

- Keywords: the word “Cassiopea” is already present in the title. Please, remove it from the keywords section and replace it with another one.

- Lines 28-29: not all stony corals, octocorals and sea anemones are symbiotic. I recommend replacing "the well-known" with "some".

- Line 43: please, replace “hot” with “warm”. Also, write the full name of C. andromeda, including authority and year of publication.

- Line 45: why is having a "sticky interface" significant? This phrase could be a little more informative.

Line 47: please, add “SOME” stony corals and specify how much heat can Cassiopea tolerate as well as max and min temperature fluctuations.

- Line 98: please, write GVC in full. The same should be applied to caption of figure 1 and 3.

- Line 117: my suggestion is to add "amplification" as well.

- Line 122: did you mean amplification?

- Line 137: great work at providing the codes and describing each command. Congrats!

- Line 139: please, write ASV in full the first time.

- Line 143: out of curiosity…Did you face any limitations in the completeness of the Silva database? Do you think that using additional databases would have provided a different outcome?

- Line 151: GVC should be written in full since it's the first time it appears in the text.

- Line 152: please, rewrite it as “34.808 taxa” to keep numbers standardized.

- Line 179: what about the other 21 specimens?

- Line 181: please, remove “GVC”.

- Line 182: please, add authority and year of publication.

- Line 183: in total, 55 specimens were sampled, but only 34 were analyzed and out of these 4 were not identified as C. xamachana. Resulting in 30 specimens. Is that correct? If so, what happened to the other 17 specimens. That is not clear. What is the actual number of analyzed specimens?

- Line 196: please, remove “(top)” and “(bottom)”.

- Line 205: please, remove “(GVC)”.

- Line 228: please write ASVs in full every time it appears for the first time in a caption.

- Line 229: displayed?*

- Caption of figures 3 and 6: please, rewrite it explaining what each abbreviation mean (HHP, CK, LW…)

- Line 243: please, invert to "location (b+c)".

- Line 245: please, invert to “GVC (b) and bell (c) alpha diversity”.

- Line 256: please, write “PCA” in full. The same to line 352.

- Line 272: please, add a single space after “of” (e.g.: of > 0.1%).

- Line 280: Cassiopea should be in italics.

- Caption of figure 6: please, explain each abbreviation and write GVC in full the first time it appears in the caption. The same should be applied to all captions (tables and figures).

- Line 288: please, correct the overspacing.

- Lines 318 - 322: why some organisms are more abundant than others depending on the area where they were sampled (e.g. bell, gvc)? Could this have anything to do with their functional role?

Line 353: “were” = where?*

- Line 360: Cotylorhiza tuberculata (Macri, 1778). Authority and year of publication should be added the first time a species name is mentioned. Please, revise all species names cited throughout the manuscript.

- Line 436: please, see author guidelines for how studies should be cited. For example, here, a comma is used after “et al.”, but the same is not applied to the citation above.

- Line 446: remove “a” before “either”.

- Line 452: if the Cassiopea’s external microbiome does not differ from the sediment microbiome, does it mean that Cassiopea does not have exclusive external microbiome?

Reviewer #4: The authors performed an interesting sampling of jellyfish species across a broad area. However, all analyses were performed by grouping the samples according to tissue type, regardless of the sampling locations. This decision makes it difficult to assess the geographical variability of the microbiome across the study area. While this does not compromise the scientific claims, it does not align with a data-oriented approach. Since microbiome sequencing is a data-driven method, in which ASVs are sequence constructs derived from the original samples, a data-driven pipeline is necessary to ensure that scientific hypotheses are tested without bias. For this reason, I cannot recommend the acceptance of this study at this stage.

Furthermore, the manuscript contains many small issues, such as typos (e.g., refraction curves – L155; Bray curtis). In other sections, some sentences are poorly constructed, such as the use of "bacteria" as an adjective in L191: “The dataset generated was predominantly bacteria with few archael reads.” This sentence also includes a typo: archaeal (not archael). A similar word choice appears to have been made in the title. Overall, the writing needs improvement.

Another issue is that there are too many figures in the main text. Some of them, such as Figure 1, could be moved to the Supplementary Files. The same applies to Table 1. Many of the remaining figures should also be improved in terms of quality and explanation, as they currently appear without adequate context. Why were the main phyla and classes depicted? Why use Euclidean distance, Faith’s PD, or calculate the core microbiome? What is the Dirichlet multinomial mixture (DMM), and why was it used?

The rationale behind assessing the substrate and water samples is unclear. Grouping eight samples (one per site) and comparing them with thirty GVC and bell samples in Figure 2 seems counterintuitive. Please explain this logic more clearly to the reader. A simple Venn diagram of shared ASVs would more effectively depict the percentage of ASVs shared among bell, water, substrate, and GVC samples. I could not extract this information from Table 2.

Still regarding Figure 2, an "unknown" category should be added to group ASVs classified only as "Bacteria." This would create a bar that fills each sample (or group of samples) to 100%, aligning the bars in the plot. It is also helpful to limit the figure to the x most abundant taxa to avoid cluttering the legend with indistinguishable shades of similar colors.

I don't think the in-text Shannon values (L234–L240), or any diversity metric, are informative as currently presented. These should be included in a Supplementary Table with appropriate measures of dispersion and statistical tests, indicating statistically distinct groups with letters.

The exclusive use of Euclidean distance in this analysis raises some concerns. Given the compositional and sparse nature of microbiome data, Bray-Curtis dissimilarity is generally more appropriate, as it better captures differences in community composition and is less affected by shared absences or large differences in total counts. It is recommended that the authors either adopt Bray-Curtis or provide a clear rationale for choosing Euclidean distance.

Why was only the GVC core microbiome analyzed and then searched in other samples (Figure 6)? What do the abbreviations GB, CK, VP, BKP, etc., refer to in this figure? These should be defined in the legend. Also, refer to the sampling table that should include these abbreviations. A map or schematic of these sites would be valuable for readers unfamiliar with the study area. I encourage the authors to consider improved ways to present this result. Figures 7 and 8 could be merged and moved to the Supplementary Material.

Regarding the DMM, no figure shows the model fit against the number of components (clusters), as stated in L309: “model probability value was optimal for two communities, one GVC and one bell.” This result should be presented to justify grouping the samples by tissue, as previously critiqued.

I cannot read the text in Figure 10. Please explain the star plot in the Results section.

The “Host medusa species identity” section should not be the final result, as it is disconnected from the rest of the section. If the intent was to show that there is no microbiome distinction between CA and CX, this could be stated at the beginning of the Results and shown in the Supplementary Material.

Methods

What is meant by “Retention metrics” (L144)? Does this refer to the number of reads retained after preprocessing? Please clarify this for the reader.

The provided GitHub repository (https://github.com/kadenmuffett/Cassiopea-Code%20Microbiome-Repository) is not publicly accessible. Either correct this or provide the code as a Supplementary File.

Why are WTR and substrate samples not described in the Sampling section alongside bell and digestive cavity samples (L103–L111)? Abbreviations used throughout the manuscript should be introduced in this section, before they appear in the Results. For example, GVC is mentioned for the first time in the Table 1 legend without prior definition.

Why is Bray-Curtis mentioned in the Methods (L161) but not in the Results? Remove it if it was not used. Please consider my suggestion above regarding this metric.

Figures 1, 2, and 7 are of poor quality — the text is unclear. Were they uploaded as vector files? In Figure 2a, the text within the box is unreadable. Also, regarding the Results, current phylum-level nomenclature should be used alongside classical names (L192–L193). For example, Pseudomonadota (formerly Proteobacteria).

While I don’t prefer sections within the Discussion, the text is interesting and connects the findings to the scientific literature. Including more explicit references to the underlying Results would further improve it.

Minor corrections

Please cite SILVA database (10.1093/nar/gks1219) (L144)

Refraction curves? (L155). Rarefaction

**Do you want your identity to be public for this peer review?** For information about this choice, including consent withdrawal, please see our Privacy Policy

Reviewer #1: No

Reviewer #3: No

Reviewer #4: No

---

## [Author Response · Author response to Decision Letter 2]

29 Apr 2025

Response to editor and reviewers:

To all,

Thank you for taking the time to review this manuscript. Your suggestions and critiques have been very helpful. The majority of changes in this round of the manuscript cover refinement of captions, movement of extraneous figures to the supplementary materials, and greater detail in some sections of the methodology section. Hopefully the in-text modifications improve the clarity and readability of the text.

Sincerely,

KMM

Reviewer 1

To Reviewer 1,

Thank you for your specific critiques. As you have suggested, the ethanol preserved samples have been removed from the methods section in order to improve clarity. Table #1 has been separated to provide adequate space for definitions and the term “GVC” is now clarified earlier. Additional analyses of the physical parameters have now been added to the supplements, as is addressed in the specific response below. Additionally, the limited nature of this sampling has now been reiterated in the core microbiome section.

Thank you,

Kaden Muffett

Specific points

Briefly, since the samples preserved in alcohol did not give good results, these should be removed from the manuscript.

The authors accept this suggestion and have modified the text accordingly.

Table 1 needs work with headings, column spaces, and some definitions.

The GVC term should be introduced in Methods.

Thank you for bringing up this oversight. The correction has been made.

The authors collected information on physical parameters, but such information was not relevant in their analyses, correct?

Yes, as these were point-in-time measurements, it was deemed imprudent to include them in the primary analysis. Redundancy analysis plots (RDA) including these features of interest have now been added to the supplementary materials for those interested. They are presented in this table to inform others’ interpretation of these data, as factors associated with these have been demonstrated important in other microbiomes.

Finally, in the Results section referring to Core Microbiome, the authors need to stress that the sampling was done in one month within a single year.

Thank you for the suggestion, this has now been specified in the results.

Reviewer #3

To reviewer #3,

Thank you for your review of this manuscript. Your suggestions have improved the precision and clarity of the text. In addition to your specific comments, you also indicated some more general inquiries regarding the nature of these microbiomes. While I am not capable of speaking to the function of the Cassiopea microbiome with these data, I do think that these questions are worth pursuing with other study designs.

Please see the responses to all of your specific requested edits below.

Sincerely,

KMM

Specific comments

Title: because both internal and external microbiomes were investigated, the words “gastric microbiome” limits the scope of the study, which is broader. My suggestion is to alter it.

Thank you for your suggestion. The title has been modified to include both external and internal data.

Keywords: the word “Cassiopea” is already present in the title. Please, remove it from the keywords section and replace it with another one.

Corrected.

Lines 28-29: not all stony corals, octocorals and sea anemones are symbiotic. I recommend replacing "the well-known" with "some".

Corrected.

Line 43: please, replace “hot” with “warm”. Also, write the full name of C. andromeda, including authority and year of publication.

Corrected.

Line 45: why is having a "sticky interface" significant? This phrase could be a little more informative.

Corrected. The text now specifies that this is a barrier between host and environment.

Line 47: please, add “SOME” stony corals and specify how much heat can Cassiopea tolerate as well as max and min temperature fluctuations.

Corrected, specific tolerance details added.

-Line 98: please, write GVC in full. The same should be applied to caption of figure 1 and 3.

Thank you, this issue has been corrected.

Line 117: my suggestion is to add "amplification" as well.

Corrected.

Line 122: did you mean amplification?

Corrected.

Line 137: great work at providing the codes and describing each command. Congrats!

Thank you for your praise.

Line 139: please, write ASV in full the first time.

Corrected.

Line 143: out of curiosity…Did you face any limitations in the completeness of the Silva database? Do you think that using additional databases would have provided a different outcome?

It is certainly possible. The Silva database has made substantial strides in the last several years when it comes to marine taxa, so issues I had with ID in the first tests (2021) are largely absent now. RDP certainly was considered, but the comparability of Silva across studies is currently unbeatable. I also considered using the expanded McCauley classifier, but decided that, should someone want perfect taxonomic comparability to another study, they will likely need to reclassify anyway. Such is the reality of amplicon sequencing, and also our constantly in-flux marine microbial genera.

Line 151: GVC should be written in full since it's the first time it appears in the text.

This has now been corrected earlier in the text (methodology section)

Line 152: please, rewrite it as “34.808 taxa” to keep numbers standardized.

Corrected.

Line 179: what about the other 21 specimens?

The other 21 included medusae under 3 cm diameter (9 indv), and samples that did not fit on the final sequencing run with the resources available. This has now been specified in the text.

Line 181: please, remove “GVC”.

Corrected.

Line 182: please, add authority and year of publication.

Corrected.

Line 183: in total, 55 specimens were sampled, but only 34 were analyzed and out of these 4 were not identified as C. xamachana. Resulting in 30 specimens. Is that correct? If so, what happened to the other 17 specimens. That is not clear. What is the actual number of analyzed specimens?

Thank you, the description of the dataset has been standardized across the paper. It is now clear that 55 were collected, of these 46 were swabbed, of these 34 samples were sequenced, and of these 30 were C. xamachana.

Line 196: please, remove “(top)” and “(bottom)”.

Corrected.

Line 205: please, remove “(GVC)”

Corrected.

Line 228: please write ASVs in full every time it appears for the first time in a caption.

Corrected.

Line 229: displayed?*

Corrected.

Caption of figures 3 and 6: please, rewrite it explaining what each abbreviation mean (HHP, CK, LW…)

Thank you for your feedback. The explanations are now available within table 1.

Line 243: please, invert to "location (b+c)". Line 245: please, invert to “GVC (b) and bell (c) alpha diversity”, Line 256: please, write “PCA” in full. The same to line 352.

Corrected.

Line 272: please, add a single space after “of” (e.g.: of > 0.1%).

Corrected.

Line 280: Cassiopea should be in italics.

Corrected.

Caption of figure 6: please, explain each abbreviation and write GVC in full the first time it appears in the caption. The same should be applied to all captions (tables and figures).

This has been corrected across captions.

Line 288: please, correct the overspacing.

Corrected.

Lines 318 - 322: why some organisms are more abundant than others depending on the area where they were sampled (e.g. bell, gvc)? Could this have anything to do with their functional role?

It is entirely possible that this is related to functional role, however the assessment of functional role from 16S data, especially from divergent taxa, is difficult and error prone. This question is outside the scope of this paper but certainly an interesting one.

Line 353: “were” = where?*

Corrected.

Line 360: Cotylorhiza tuberculata (Macri, 1778). Authority and year of publication should be added the first time a species name is mentioned. Please, revise all species names cited throughout the manuscript.

Corrected.

- Line 436: please, see author guidelines for how studies should be cited. For example, here, a comma is used after “et al.”, but the same is not applied to the citation above.

The erroneous citations have been modified, thank you for bringing this to my attention.

Line 446: remove “a” before “either”.

Corrected.

Line 452: if the Cassiopea’s external microbiome does not differ from the sediment microbiome, does it mean that Cassiopea does not have exclusive external microbiome?

This is a reasonable interpretation of the data. This community is enriched in Endozoicommonas relative to the benthos, as discussed elsewhere, but aside from this, the community is very similar to that of surrounding substrates.

Reviewer #4

Reviewer 4,

Thank you for taking the time to review this manuscript. You have raised two main concerns about the structure of this paper, the first being that we are not fully able to parse the impact of location with the samples collected, and secondly, that this paper has lapses in explanations of methodological purpose.

To the first point, I agree that the sample sizes are not adequate across all sites to justify this being the driver of the paper. In reality, what we have the ability to analyze is only the ways in which gastrovascular cavities and bells show consistencies. In order to bring the introduction into line with this, the introduction has been modified, including both stressing that this is an exploratory paper and that the focus is to identify the core components of the GVC and bell communities.

To the second point, I have modified the paper such that the impetus for specific tests are clearer, and switched out the less informative DMM figure with a more informative one. Where there were too many figures, they have been moved to the supplementary materials, and where there was inadequate detail in the supplements (eg Shannon diversity), the supplementary materials have been expanded.

In addition to these, in alignment with your suggestion, Bray-Curtis is now used throughout the text.

Thank you for your feedback,

Kaden Muffett

Specific comments:

Since microbiome sequencing is a data-driven method, in which ASVs are sequence constructs derived from the original samples, a data-driven pipeline is necessary to ensure that scientific hypotheses are tested without bias.

Thank you for this criticism. In this case, this is an exploratory study of this community, and while I understand the concern for bias, I believe there to be value in the data collected when framed as variability within the Cassiopea microbiome. I accept your criticism as this: The aim of this paper specifically states “compare the microbial communities of Cassiopea across sites of the Florida Keys”. I agree that I do not have adequate replicates to fully elucidate differences between sites. As such, I have changed the wording of this aim to reflect the core of what this work has accomplished, “The aim of this work is to identify the core components of the internal and external mucosal microbiomes of Cassiopea within the Florida Keys.”.

This sentence also includes a typo: archaeal (not archael). A similar word choice appears to have been made in the title. Overall, the writing needs improvement.

The typos throughout the text have been corrected.

Another issue is that there are too many figures in the main text. Some of them, such as Figure 1, could be moved to the Supplementary Files. The same applies to Table 1.

Thank you for your suggestion. While I understand that some of these figures and tables may seem extraneous, this paper is serving a diverse group of scientists, and other reviewers have been and remain interested in the inclusion of this table.

Many of the remaining figures should also be improved in terms of quality and explanation, as they currently appear without adequate context.

Thank you for this criticism. The figure captions have been modified to better explain the contents of each figure.

Why were the main phyla and classes depicted?

The main phyla and classes are depicted in this paper because of precedent. For past coral (Dunphy et al 2019 (https://doi.org/10.1038/s41598-019-43268-6), Hernandez-Agreda 2018 (https://doi.org/10.1128/mbio.00812-18)) and scyphozoan microbiome work (Ohdera et al 2024 (https://doi.org/10.1371/journal.pone.0298002) , Peng et al 2021 (https://doi.org/10.3389/fmicb.2021.647089)), this has been a common display mechanism. This figure is displayed for easy communication of the major groups found in these samples within this study.

Why use Euclidean distance, Faith’s PD, or calculate the core microbiome?

Thank you for your questions. In this case, I did not fully consider the impact of Euclidean diversity. I have changed all beta diversity to Bray-Curtis throughout the text. For Faith’s PD, I have now specified the following in the methods section “In order to identify the degree to which diversity spanned primarily shallow or deep phylogenetic distances, Faith’s phylogenetic diversity was calculated for all samples and sample types. Faith’s phylogenetic diversity was computed..” . For the core microbiome, an explanation of choice is now provided in the methods “In order to identify taxa consistently present across samples of a given type that may be rare, core microbiome was calculated for internal and external communities of Cassiopea. Core microbiome was computed..”

What is the Dirichlet multinomial mixture (DMM), and why was it used?

Dirichilet multinomial mixture analysis tests an ideal number of populations for a provided diverse population set. The purpose of this has now been clarified in the methods.

“In order to identify whether a gastrovascular/ bell split community partitioning was appropriate for all samples, and identify subcommunity level enterotypes within Cassiopea internal and external microbiomes, Dirichlet Multinomial mixture analysis, as described by Holmes and implemented in the DirichletMultinomial package v 1.48 was used for Cassiopea – derived samples (52,53). “

The rationale behind assessing the substrate and water samples is unclear. Grouping eight samples (one per site) and comparing them with thirty GVC and bell samples in Figure 2 seems counterintuitive. Please explain this logic more clearly to the reader. A simple Venn diagram of shared ASVs would more effectively depict the percentage of ASVs shared among bell, water, substrate, and GVC samples. I could not extract this information from Table 2.

The inclusion of the substrate and water in figure 2 and table 2 serve as comparison tools to the microbiomes discussed. An upset plot of group to group overlap is now included.

Still regarding Figure 2, an "unknown" category should be added to group ASVs classified only as "Bacteria." This would create a bar that fills each sample (or group of samples) to 100%, aligning the bars in the plot. It is also helpful to limit the figure to the x most abundant taxa to avoid cluttering the legend with indistinguishable shades of similar colors.

Thank you for your suggestion. The colors and design for this have been altered as suggested to aid comprehensibility.

I don't think the in-text Shannon values (L234–L240), or any diversity metric, are informative as currently presented. These should be included in a Supplementary Table with appropriate measures of dispersion and statistical tests, indicating statistically distinct groups with letters.

Thank you for the suggestion. A table of Shannon diversity metrics is now available in the supplementary materials.

Why was only the GVC core microbiome analyzed and then searched in other samples (Figure 6)? What do the abbreviations GB, CK, VP, BKP, etc., refer to in this figure? These should be defined in the legend. Also, refer to the sampling table that should include these abbreviations. A map or schematic of these sites would be valuable for readers unfamiliar with the study area. I encourage the authors to consider improved ways to present this result. Figures 7 and 8 could be merged and moved to the Supplementary Material.

The specificity of this gastrovascular cavity is of interest to the field. The explicit separation of internal and external samples is rarely done for cnidarian samples. As such, making it clear that the core of this community is not represented in other sam

---

## [Decision Letter · Decision Letter 2]

15 Jul 2025

Dear Dr. Muffett,

Thank you for submitting your manuscript to PLOS ONE. After careful consideration, we feel that it has merit but does not fully meet PLOS ONE’s publication criteria as it currently stands. Therefore, we invite you to submit a revised version of the manuscript that addresses the points raised during the review process.

Please note that all three reviewers had a few *very* minor remaining points. PlosONE policy requires that these minor changes are taken care of before the manuscript can be accepted for publication.

Statements in abstract: The high diversity of bell-associated microbes should be mentioned. Figure 2: The correct naming convention for Actinobacteria should be double checked (Actinobacteria vs. Actinomycetota).Figure 9 requires some more clarification regarding its description, in particular the term ‘identical’, axis labeling, and description of specific sections in the figure. This should be clarified.Please note that one of the reviewers has uploaded an annotated PDF highlighting a few remaining points.Editor’s note: Please also double check the correct spelling for microbial taxa. ‘Endozoicomonas’ should be spelled with only 1 ‘m’. There are 14 instances of misspelling of this genus name (with 2 ‘m’, i.e., ‘Endozoicommonas’), including in the title. 

Below are the more detailed requests by the expert reviewers. 

We look forward to receiving your revised manuscript.

Kind regards,

Claudia Isabella Pogoreutz

Academic Editor

PLOS ONE

Journal Requirements:

Reviewers' comments:

Reviewer's Responses to Questions

**Comments to the Author**

Reviewer #1: All comments have been addressed

Reviewer #3: All comments have been addressed

Reviewer #4: (No Response)

2. Is the manuscript technically sound, and do the data support the conclusions?

Reviewer #1: Yes

Reviewer #3: Yes

Reviewer #4: Yes

3. Has the statistical analysis been performed appropriately and rigorously?

Reviewer #1: Yes

Reviewer #3: I Don't Know

Reviewer #4: Yes

4. Have the authors made all data underlying the findings in their manuscript fully available?

Reviewer #1: Yes

Reviewer #3: Yes

Reviewer #4: Yes

5. Is the manuscript presented in an intelligible fashion and written in standard English?

Reviewer #1: Yes

Reviewer #3: Yes

Reviewer #4: (No Response)

Reviewer #1: In the abstract, I find the high diversity of the bell-associated microbes, with no clear dominance of any group, more relevant than the written phrase “Cassiopea bell mucosal samples conform largely to the communities of surrounding sediment with the addition of Endozoicomonas cf. atrinae.” Perhaps the authors can include both?

Minor recomendations:

Lines 24 and 449: Symbiodiniaceae should NOT be in italics

Line 38: in the parenthesis, Symbiodiniaceae is not a clade, is a family

Reviewer #3: The authors addressed all my comments and made clear improvements in this new version. There are still minor errors that should be corrected (PDF attached), but after that I think it is good to go.

Congratulations on the work!

Reviewer #4: The manuscript has been significantly improved. Although it still lacks a clearly stated hypothesis, it successfully meets its aim of conducting a preliminary study on the bacterial microbiomes of these marine organisms.

Figure 9 and its accompanying description remain somewhat unclear. I found the use of the term "identical" (L379) confusing, as its meaning is only clarified later in the Discussion section (L531). I recommend that the BLAST search, in which an ASV is used as a query, be described explicitly in the Results section.

Regarding Figure 9, the y-axis label is "Abundance," yet the authors mention that the abundance reached "up to 10% within the sample" (L424). In this context, the y-axis label should be revised to "Relative abundance." Additionally, if only Vibrio coralliilyticus reads were plotted, the figure’s appearance as a stacked bar plot is misleading. What do the sections within each bar represent—values from samples of the same type or location? The bars are also visually cluttered; replacing the inner background grid lines with area shading or patterned fills could improve readability.

Figure 2 – Actinobacteriota. Does it refer to Actinobacteria? If it does, the accepted nomenclature is Actinomycetota. Please consult the LPSN website to use the up-to-date nomenclature (https://lpsn.dsmz.de/).

Minor corrections:

"Spoke" (L411) appears out of context—please clarify or revise.

Figure 9’s color legend should include the genus name or abbreviation for clarity.

what does this mean?

**Do you want your identity to be public for this peer review?** For information about this choice, including consent withdrawal, please see our Privacy Policy

Reviewer #1: No

Reviewer #3: **Yes: ** Hellen Ceriello

Reviewer #4: No

---

## [Author Response · Author response to Decision Letter 3]

22 Jul 2025

Reviewer response

Dear editor and reviewers,

Thank you for your thoughtful commentary and criticism. In response to these reviews, figures 2 and 9 have been corrected, and a variety of clarity and spelling issues have been fixed throughout the text. I hope that these changes are satisfactory.

Sincerely,

KMM

To the editor;

The following changes have been made in line with suggestions:

1) Statements in abstract: The high diversity of bell-associated microbes should be mentioned.

We have accepted this note and modified the abstract.

2) Figure 2: The correct naming convention for Actinobacteria should be double checked (Actinobacteria vs. Actinomycetota).

The outdated name has been replaced with the correct “Actinomycetota” within the figure.

3) Figure 9 requires some more clarification regarding its description, in particular the term ‘identical’, axis labeling, and description of specific sections in the figure. This should be clarified.

For figure 9, the y-axis, title and figure label have been modified to impart greater clarity. “Identical” has been removed not all are identical within the NCBI rRNA database. A supplementary table has been added with best matches for the top six Vibrio cf. coralliilyticus sequences and referenced within the text in both the methods and results.

4) Please note that one of the reviewers has uploaded an annotated PDF highlighting a few remaining points.

The typo fixes and clarifications provided by reviewer two have been accepted into the body of the text.

5) Editor’s note: Please also double check the correct spelling for microbial taxa. ‘Endozoicomonas’ should be spelled with only 1 ‘m’. There are 14 instances of misspelling of this genus name (with 2 ‘m’, i.e., ‘Endozoicommonas’), including in the title.

Corrected.

Comments to the Author

Reviewer #1

To reviewer #1,

Thank you for reviewing. Your suggestion and minor recommendations have been accepted into the text.

Sincerely,

Kaden Muffett

1) In the abstract, I find the high diversity of the bell-associated microbes, with no clear dominance of any group, more relevant than the written phrase “Cassiopea bell mucosal samples conform largely to the communities of surrounding sediment with the addition of Endozoicomonas cf. atrinae.” Perhaps the authors can include both?

Corrected.

2) Minor recomendations:

Lines 24 and 449: Symbiodiniaceae should NOT be in italics

Line 38: in the parenthesis, Symbiodiniaceae is not a clade, is a family

Corrected.

Reviewer #3

To reviewer #3,

Thank you for reviewing, as well as for your praise of the manuscript. Your edits identifying typos and grammatical errors has significantly improved the readability of the text.

Sincerely,

Kaden Muffett

Reviewer #4

To reviewer #4,

Thank you for reviewing. Figures 9 and 2 have been revised to correct identities and improve clarity. The word “Spoke” has been replaced with “point” as readers are likely more familiar with this in star plots. Details of the ASV blast search are now included in the text.

Sincerely,

Kaden Muffett

1) Figure 9 and its accompanying description remain somewhat unclear. I found the use of the term "identical" (L379) confusing, as its meaning is only clarified later in the Discussion section (L531). I recommend that the BLAST search, in which an ASV is used as a query, be described explicitly in the Results section. Regarding Figure 9, the y-axis label is "Abundance," yet the authors mention that the abundance reached "up to 10% within the sample" (L424). In this context, the y-axis label should be revised to "Relative abundance." Additionally, if only Vibrio coralliilyticus reads were plotted, the figure’s appearance as a stacked bar plot is misleading. What do the sections within each bar represent—values from samples of the same type or location? The bars are also visually cluttered; replacing the inner background grid lines with area shading or patterned fills could improve readability.

The blast search is now described in the text and present in a supplementary table with closest blast identities from the curated NCBI rRNA database. The y-axis has been changed to relative abundance, the title has been clarified, as has the description. The uninformative legend was removed and the x-axis was modified to improve readability.

2) Figure 2 – Actinobacteriota. Does it refer to Actinobacteria? If it does, the accepted nomenclature is Actinomycetota. Please consult the LPSN website to use the up-to-date nomenclature (https://lpsn.dsmz.de/).

Thank you for the feedback. This has now been corrected.

3) Minor corrections:

"Spoke" (L411) appears out of context—please clarify or revise.

“Spoke” has been replaced with the more common “point” for the starplot.

---

## [Editor Report · Decision Letter 3]

29 Jul 2025

Florida Keys Cassiopea host benthos-like external microbiomes and a gut dominated by Vibrio, Endozoicommonas and Mycoplasma

PONE-D-24-22287R3

Dear Dr. Muffett,

We’re pleased to inform you that your manuscript has been judged scientifically suitable for publication and will be formally accepted for publication once it meets all outstanding technical requirements. 

Kind regards,

Claudia Isabella Pogoreutz

Academic Editor

PLOS ONE

Additional Editor Comments (optional):

Congratulations!